# Relation-Oriented: Toward Causal Knowledge-Aligned AGI

## Abstract

*Observation-Oriented* paradigm currently dominates relationship learning models, including AI-based ones, which inherently do not account for relationships with temporally nonlinear effects. Instead, this paradigm simplifies "temporal dimension" to be a *linear observational* timeline, necessitating the prior identification of effects with specific timestamps. Such constraints lead to *identifiability difficulties* for dynamical effects, thereby overlooking the potentially crucial temporal nonlinearity of the modeled relationship. Moreover, the *multi-dimensional* nature of Temporal Feature Space is largely disregarded, introducing *inherent biases* that seriously compromise the robustness and generalizability of relationship models. This limitation is particularly pronounced in large AI-based causal applications.

Examining these issues through the lens of a *dimensionality framework*, a fundamental misalignment is identified between our *relation*-indexing comprehension of knowledge and the current modeling paradigm. To address this, a new *Relation-Oriented* paradigm is raised, aimed at facilitating the development of causal knowledge-aligned Artificial General Intelligence (AGI). As its methodological counterpart, the proposed *Relation-Indexed Representation Learning* (RIRL) is validated through efficacy experiments.

## 1 Introduction

The current modeling paradigm requires prior identification of variables and outcomes as a prerequisite for constructing the relationship over them, typically based on observational independent and identical distributions (i.i.d.). With respect to the time evolution of these i.i.d.s, the Picard-Lindelof theorem, introduced in the 1890s, established a *logical timeline t* for recording observational timestamps, thereby initiating the $x_{t+1} = f(x_t)$ paradigm to depict the time evolution of variable $X$. Since then, this **Observation-Oriented** principle has been a conventional approach to relationship learning.

To model a causal relationship $X \rightarrow Y$, the AI-based RNN models act as state-of-the-art Shojaie & Fox (2022), especially for capturing nonlinear features of the causes. They typically formulate as $y_{t+m} = f(\{x_t\})$, where $\{x_t\} = \{x_1, \ldots, x_t, x_{t+1}, \ldots, x_T\}$ denotes a time sequence of $X$ of length $T$, with a predetermined time progress $m$ from $X$ to $Y$. In this approach, the temporal distribution of cause $X$ is explicitly included, while outcome $Y$ strictly presents as observational, associated with a specific timestamp. This way leaves all potentially significant dynamics of effects entirely managed by $f(\cdot)$. However, whether the selected function $f(\cdot)$ is *linear* or *nonlinear* influences only the dimensionality of $\mathbb{R}^d$, where $X \in \mathbb{R}^d$. Consequently, the time evolution from $t$ to $t + m$ for the effect entity $Y$ remains invariably **linear**.

Not due to specific models, such limitation on capturing temporal nonlinearities results from the prevailing *Observation-Oriented* paradigm. Specifically, before modeling the relationship, it requires manual identification of the effect in specific timestamps, thus posing difficulties when the effect presents diverse dynamical features Zhang (2012). While the paradigm may have been adequate in the past, it no longer suffices given the advancements in data collection and Artificial Intelligence (AI) methods. The reliance on i.i.d. observations, coupled with the growing necessity for capturing **dynamics** (i.e., *temporal* **nonlinearities** Granger (1993)), underscores the need for a new modeling paradigm Scholkopf (2021).

Drawing inspiration from the relation-centric nature of human comprehension Pitt (2022), this study presents a *dimensionality framework*. This offers a renewed perspective on the concept of "relationship" within the modeling context, underscoring the vital **indexing** role of unobservable relations in capturing observable entities, especially temporal linearities. The unique viewpoint uncovers a fundamental misalignment between our intuitive understanding of knowledge and the prevailing relationship learning paradigm, resulting in

*inherent biases* within models. This issue plays a significant role in several challenges, such as the difficulty of generalizing causal models Scholkopf (2021), the limited effectiveness in leveraging causal knowledge within AI Luo (2020), and certain phenomena associated with AI Alignment problems Christian (2020).

The remainder of this Introduction (subsection 1.1-1.3) lays the groundwork for the *dimensionality framework* used throughout this study. Chapter I (Sections 2-4) examines the inherent limitations of *Observation-Oriented* relationship learning, particularly its oversight of multi-dimensional dynamics in causal effects. The chapter also introduces the **Relation-Oriented** paradigm, which is crafted to reflect human understanding. Chapter II (Sections 5-7) concentrates on the *Relation-Indexed Representation Learning* (RIRL) method as a practical realization of the proposed new paradigm, accompanied by efficacy experiments.

## 1.1 Dimensionality Framework

Consider a pairwise relationship comprised of three elements: two **observable** entities, and a relation derived from our knowledge to connect them. These two entities can be featured as observational only (e.g., images, spatial coordinates of a quadrotor, etc.), or observational-and-temporal (e.g., trends of stocks, a quadrotor's movement in one hour, etc.). However, the "relation" has to be **unobservable** to make this relationship model *informative*, to be distinguished from mere statistical dependency between two observables.

This principle was initially introduced in the form of Common Cause Dawid (1979); Scholkopf (2021), suggesting that any nontrivial conditional independence between two observables requires a third, mutual cause (i.e., our unobservable "relation"). Take the relationship "Bob has a son named Jim" as an example. The father-son relation is unobservable information that exists in our knowledge, which can also be seen as the common cause that makes their connection unique rather than any random pairing of "Bob" and "Jim". Given sufficiently observed social activities, AI may deduce this pair of "Bob" and "Jim" are particularly associated, but that does not equate to discerning the father-son relation between them.

Put simply, the information contained by a relationship model stems from unobservable knowledge (referred to as "relation") rather than associated direct observations. Consider model $Y = f(X; \theta)$ with $\theta$ indicating the function parameter in demand. In the context of modeling, the term "relation" can be represented by $\theta$.

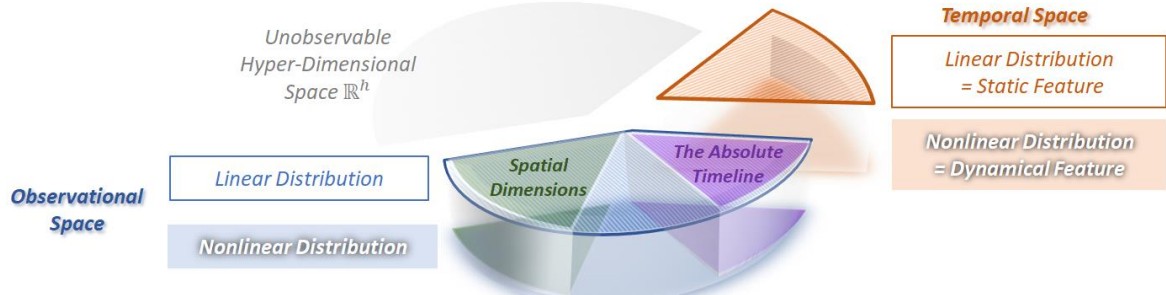

Figure 1: Dimensionality Framework: splitting the Knowledge-Storing Cognitive Space according to the features accommodated, i.e., Observational, Temporal, and Hyper-dimensional Feature Spaces.

Therefore, in modeling, a relationship can be interpreted as a joint distribution spanning multiple dimensions. The observational and temporal dimensions include the entities (i.e., $X$ and $Y$), while the unobservable relation (i.e., the modeling objective $\theta$) manifests as an unseen distribution in a **hyper-dimension**. Figure 1 organizes our cognitive space, which stores knowledge, into three sections accordingly. The hyper-dimensional space represents the aggregate of all unobservable relations in our knowledge. For a model to be practically valuable, it must accurately reflect our understanding. Similarly, a successful AGI, to meet our expectations, must be rooted in existing knowledge. Specifically, it should represent relations residing in the unobservable Hyper-dimensional Space, through which reasonable interpretations of observable entities can be generated.

In this paper, "feature" refers to a variable fully representing a distribution of interest in any dimension, while the observational-temporal joint space is sometimes called "observable data space", contrasting with "latent feature space".

## 1.2 Observational and Temporal Feature Spaces

Most relationship models function within the Observational Space, maybe incorporating a timeline to depict the observational evolution over time. For example, Convolutional Neural Networks (CNNs) recognize pixel associations in purely observational space; a quadrotor's movement is identifiable in a sequence of spatial coordinates; Large Language Models (LLMs) operate along a semantic timeline representing phrase order; and patients' vital signs are recorded chronologically. The latter three applications fall under the category of "spatial-temporal" analysis Alkon (1988); Turner (1990); Andrienko (2003), where the "temporal dimension" is often equated with the observational timeline within the data Wes (2023).

However, our cognitive understanding of "time", serving as the foundation to construct knowledge, differs from this approach Coulson (2009). Observational data timestamps are referred to as the **absolute** timeline Wulf (1994), while in comprehension, *multiple* **relative** timelines can coexist. Each of these relative timelines represents different causal effects and may exert mutual influences Shea (2001). Additionally, from a modeling perspective, data timestamps are indistinguishable from other observational attributes Shea (2001). Consequently, we categorize the *absolute t*-timeline as a dimension in the Observational Space (as depicted in Figure 1); meanwhile, address knowledge-aligned *temporal distributions* separately in a distinct Temporal Space, which naturally possesses **multi-dimensions**, defined by the potential *relative* timelines present.

A *linear* causal relationship implies a **static** effect that can be specified by a particular timestamp, e.g., in the statement "rain leads to wet floors," the effect of "wet floors" is static, captured at a specific moment. When this effect has significant **dynamical** features - e.g., "floors becoming progressively wetter" is dynamic due to its sequential temporal pattern - a temporal distribution must be considered, transforming the relationship into a temporally *nonlinear* one Granger (1993).

Under the *Observation-Oriented* paradigm, prior identification of effects for dynamics is notably difficult (see subsection 3.2 for further discussions). This neglect of temporal nonlinearity and oversight of relative timelines can lead to **inherent bias** (as demonstrated in subsection 4.1), thereby compromising the generalizability of causal models (see subsection 4.2). While such misalignments might have been subtle in the past, the advent of AI enables large-scale models more efficiently, and its black-box nature allows these biases to accumulate exponentially inside, eventually resulting in uninterpretable outputs.

## 1.3 Hyper-Dimensional Feature Space

In Hyper-dimensional Space (denoted as $\mathbb{R}^h$), unobservable relations include not only the modeling objective $\theta$, but also other ones that play crucial roles for the model. Consider $\langle \theta, \omega \rangle$ to be jointly distributed in $\mathbb{R}^h$, connecting observables $X$ and $Y$. While the model $Y = f(X; \theta)$ aims to obtain $\theta$ using given $X$ and $Y$, the unseen $\omega$ can imply various application scenarios that necessitate the model's **generalizability**.

For instance, consider an examination of how family income levels (denoted as $X$) influence grocery shopping frequencies (as $Y$), with influence represented by $\theta$. Underlying cultural factors (denoted as $\omega$) also play a role, such that the established model $Y = f(X; \theta)$ proves to be practically useful only when conditioned on a specific country (represented by a particular $\omega$ value). In this context, there are two levels of objective relation: a global-level $\theta$ without considering $\omega$, and a local-level $\theta$ conditioned on a specified $\omega$ value.

To be *generalizable* is to traverse these levels effectively, thereby allowing lower-level learned relationships to inform or be reusable for higher-level learnings Scholkopf (2021). This also encompasses the capability to *individualize* inversely from higher to lower levels for different $\omega$ values. For simplicity, we refer to $\omega$ as the **hidden relation** and the resulting unseen levels as the **unobservable hierarchy**.

## Chapter I: Limitations of Current Observation-Oriented Paradigm

Human understanding inherently indexes through relations Pitt (2022), directing to mental representations about observational and temporal entities. This intrinsic characteristic results in a fundamental misalignment with the *Observation-Oriented* modeling paradigm, evident through various application issues.

Section 2 explores the impacts of hidden relations on relationship learning and introduces the relation-indexing approach as a solution. Section 3 underscores the importance of effect dynamics and the challenges of

manual identification in causal learning. Finally, Section 4 highlights the profound implications of overlooking multi-dimensional effect dynamics.

## 2 Impact of Hidden Relations

Hidden relations imply the existence of unobservable hierarchies. In strictly observational learning tasks, features across various levels can be fully captured, setting off the hidden relations as distinct observable misalignment (subsection 2.1). However, in relationship learning tasks characterized by temporal dimensions, the complete range of temporal dynamics can hardly be covered (subsection 2.2), leading to observable information loss and increased complexity in causal learning (subsection 2.3).

### 2.1 On Observational Learning

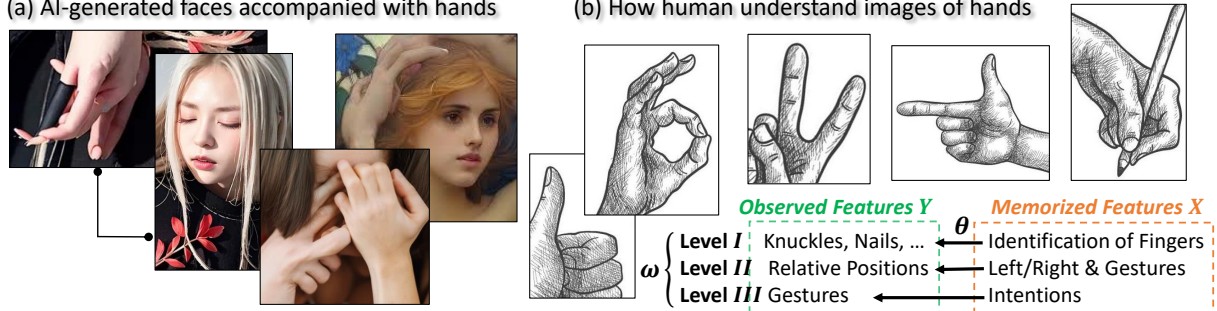

Figure 2: AI associates observational features, thus treating hands as arbitrary mixtures of finger-like items. Humans process hierarchically, indexed by relations: higher-level recognition relies on lower-level conclusions.

Figure 2(a) displays AI-generated hands with faithful colors but unrealistic shapes, while humans easily recognize plausible hands from grayscale sketches in (b). Indeed, humans hierarchically decide based on knowledge (represented as augmented feature vector $\omega = \langle \omega_1, \omega_2, \omega_3 \rangle$): **I** identifies fingers (set $\omega_1$ value); **II** discerns gestures by finger positions (set $\omega_2$ value given $\omega_1$); **III** retrieves gesture meanings (set $\omega_3$ value given $\omega_1, \omega_2$). However, the hierarchy information $\omega$ in our cognition is unseen to AI. Without guidance from the indexing relations at each level (denoted by $\theta = \{\theta_1, \theta_2, \theta_3\}$), AI discerns only associations, resulting in basic dependencies between levels of entities (e.g., $\mathbf{P}(Y_2 \mid Y_1)$) devoid of knowledgable insights (no $\omega, \theta$).

In associational learning tasks (concerning $Y$ only), the hidden $\omega$ is not always essential. If entities across levels are observationally distinct and non-overlapping, AI can accurately differentiate them. For instance, AI can generate convincing faces because the appearance of eyes strongly indicates facial angle, removing the need to distinguish "eyes" ($Y_2$) from "faces" ($Y_1$). When all observational levels are fully captured, AI can uncover the hidden $\omega$ using methods such as inverse reinforcement learning Sutton (2018); Arora (2021). For example, approvals of generated five-fingered hands may lead AI to identify fingers autonomously.

### 2.2 On Temporal Relationship Learning

Figure 3(a) shows an example from health informatics, depicting the causality from action $do(A)$ to sequence $\{B_t\}$, denoted as $\mathcal{B}$. Then, $\mathcal{B}$ can be disentangled as two levels of dynamical features: **I** the standard sequence of length 30 ($do(A) \xrightarrow{\theta_1} \mathcal{B}_1$ set $\omega = \varnothing$ ); **II** individualized progress variation ($E \xrightarrow{\theta_2} \mathcal{B}_2$ set $\omega = P_i, P_j, \ldots$), where the patient's personal characteristics $E$ is hidden. For simplicity, assume influence $\theta_2$ as linear, i.e., $E$ uniformly accelerates or decelerates the effective progress, and $\mathcal{B}_2$ simply interprets the individualized speed for patients ($\omega = P_i, P_j, \ldots$). The modeling objective is to obtain $\mathcal{B}_1$, as the effectiveness evaluation of $M_A$.

Conventionally, the clinical effectiveness of $M_A$'s is estimated by averaging the performances of all patients after 30 days, resulting in a correlation model $B_{t+30} = f(do(A_t))$. It only captures the *static* feature $B_{t+30}$, the final step of level **I** dynamic, neglecting the preceding 29 steps, as represented in Figure 3 (b).

Significantly, even when adopting a sequence to represent $\mathcal{B}_1$, as in the Granger causality model Granger (1993), capturing the level **I** dynamic through such a "sequential static" variable remains challenging (refer to subsection 3.2 for more discussions). To illustrate, obtaining an accurate estimation by averaging sequences

from D1 to D30 for all patients necessitates meeting certain criteria: an exact 30-day span; near-linear variations among individuals; and a normal distribution centered on D30; .... In essence, this method involves manually defining the boundary of $\theta_1$ by exploring all possible $\omega$ values.

Hierarchical dynamical effects are commonly observed in various applications, including epidemic progression, economic fluctuations, strategic decision-making, and so on. Traditional approaches to these challenges typically involve a manual specification regarding the potential value of $\omega$, to delimit a particular level of $\theta$. A typical example is the group-specific learning methodologies Fuller et al. (2007).

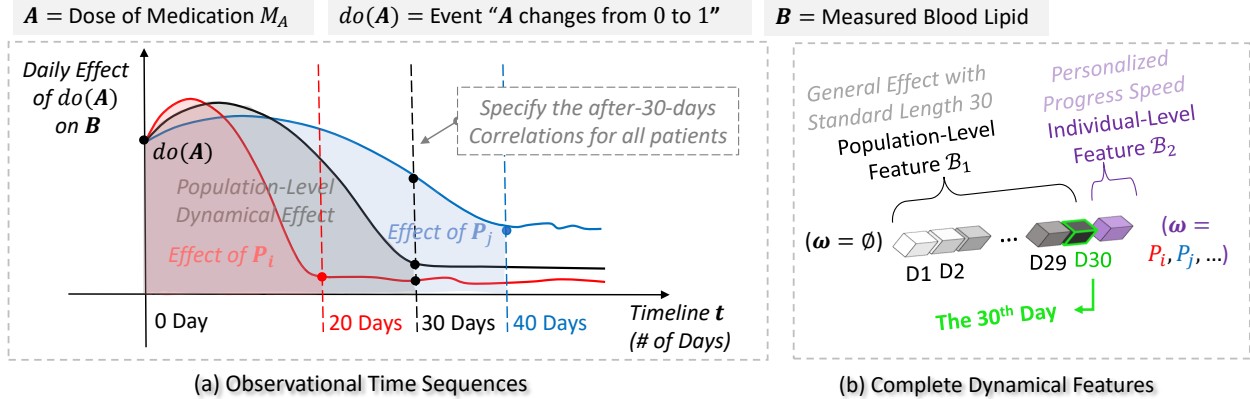

(a) Observational Time Sequences        (b) Complete Dynamical Features

Figure 3: Medication $M_A$ treats high blood lipid, with $do(A)$ denoting its initial use. It is given that the population-level effect takes about 30 days to fully release ($t = 30$ at the elbow), depicted by the black curve in (a). Patient $P_i$ achieves this effect curve elbow in 20 days, while $P_j$ takes 40 days.

## 2.3 The Elusive Hidden-Confounder

For patients $P_i$ and $P_j$, the population-level last-day effect $B_{t+30}$ is inaccurate. To counter this individual-level bias and improve model interpretation, statistical causal inference incorporates the "hidden confounder" concept into Directed Acyclic Graphs (DAG), as node $E$ in Figure 4 (a). However, this approach does not necessitate collecting additional data for $E$, leading to an illogical implication: "The model bias stems from unknown factors we don't intend to explore." This strategy indeed compensates for the overlooked level **II** dynamic, which essentially transforms an *observable* dynamical feature of the *effect* into a *hidden* observational variable, $E$, associated with the *cause* $do(A)$.

As shown in Figure 4(b), the hidden associated cause $do(A)*E$ does not offer a modelable relationship to learn $\{\theta_1, \theta_2\}$. That is, while introducing $E$ enhances the interpretation, it does not certainly improve the model to encompass further levels. Contrarily, a *Relation-Oriented* approach only treats relations $\{\theta_1, \theta_2\}$ as indices without additional modeling requirements. It allows AI to autonomously extract dynamical representations for multi-levels, with the indices being any observed identifier for $\omega$, like a patient ID, as shown in Figure 4(c). Such hierarchical disentanglement is driven by knowledge, thereby enhancing model generalizability.

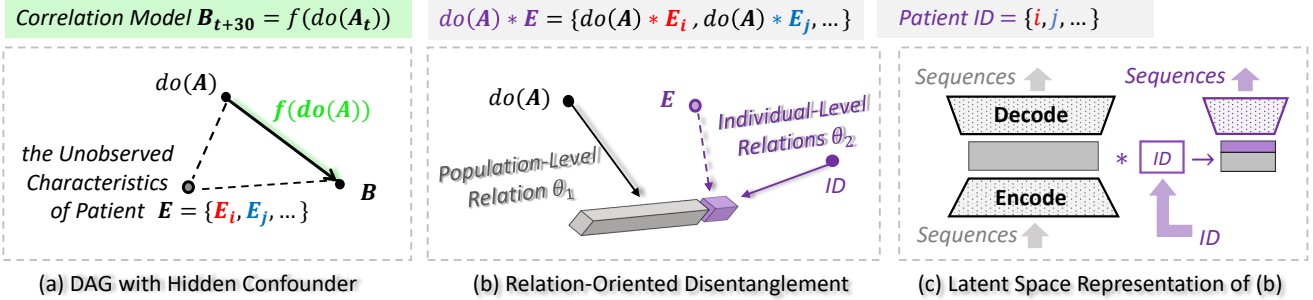

(a) DAG with Hidden Confounder    (b) Relation-Oriented Disentanglement    (c) Latent Space Representation of (b)

Figure 4: (a) Traditional causal inference DAG. (b) Hierarchical disentanglement of dynamics using relations as indices. (c) Autoencoder-based generalized and individualized reconstructions of the sequential data.

# 3  Causality on Temporal Dimension

Causal learning serves as a gateway to access the distributions within the temporal dimension, extending beyond the observational space. Under the prevailing *Observation-Oriented* paradigm, timestamps for both causal and effectual events necessitate prior specifications. This approach diverges from our instinctive understanding, where effects are identified by causes indexing through the objective causal relation.

Furthermore, timestamp specification relies exclusively on the absolute timeline, functioning merely as a regular observational dimension within the modeling context. This approach, to a degree, diminishes the temporal significance of causal relationships, rendering them indistinguishable from correlational ones from a modeling perspective, thus necessitating reliance on interpretations for differentiation.

In response, this section reexamines causality from a frequently overlooked angle - learning dynamical features of effects - with the goal of offering more intuitive insights into relevant theories and concepts. Subsection 3.1 revisits the definition of causality within the modeling context. Subsequently, subsection 3.2 distinguishes between dynamical and static variables to elucidate the challenges in effect identification. Finally, subsection 3.3 explores the limitations present in current applications of causal learning.

## 3.1  Causality in Modeling Context

Traditional causal inference often highlights model interpretations, notably distinguishing them from mere correlations, as these distinctions are not inherently embedded within the modeling context. Essentially, causality mandates the incorporation of the timeline as a *computational dimension*, ensuring recognition of significant *distributions* on it, ones that undeniably can exhibit *temporal nonlinearity*.

*Observation-Oriented* modeling fades out the causal significance of these relationships in two aspects: First, manual specifications cannot completely identify dynamics of effects for each level; Second, these dynamics might coexist in various relative timelines, which suggests multiple computational dimensions in the Temporal Feature Space. Considering these points, we revisit the definition of causality from a modeling perspective:

> **Definition 1.**  Causality vs. Correlation in the modeling context.
> - Causality = related Observational-Temporal features, including ***multi-dimensional dynamical*** *ones*.
> - Correlation = related Observational features that are ***not dynamical***.

In particular, causal modeling is vital because it facilitates the answering of *counterfactual* questions Scholkopf (2021), such as, "What effect would ensue if the cause were altered?" This capability is akin to fully capturing *temporal dimensional distributions* (i.e., all possible outcomes), thereby providing accurate responses to conditional queries (i.e., "what if" scenarios).

> **Remark 1.**  Counterfactuals can be viewed as posterior distributions in the Temporal Feature Space.

In modeling, the directionality of the relationship (i.e., the roles of cause and effect, or the "causal direction") may not impose restrictions, even though it proves important in model interpretations. Specifically, when selecting a model for the relationship $X \rightarrow Y$, one could use $Y = f(X; \theta)$ to predict the effect $Y$, or $X = g(Y; \phi)$ to inversely infer the cause $X$. Both parameters, $\theta$ and $\phi$, are obtained from the joint probability $\mathbf{P}(X, Y)$ without imposing modeling constraints. We refer to it as ***symmetric directionality*** for clarity.

In practice, concern for directionality mainly arises for two reasons: First, to maintain alignment with our intuitive understanding of temporal progression; Second, while the current paradigm can facilitate dynamical variables for the *cause*, it does not do so for the *effect* - A typical example is the RNN models.

## 3.2  Difficulty of Identifying Dynamical Effects

It is crucial to note that using a sequential variable does not necessarily capture the nonlinearity of the represented entities. The distinction between "a *sequence of **static** variables*" and "a ***dynamical*** variable" hinges on the model's ability to feature the *nonlinearity* among the sequence's elements.

RNN models technically address the challenges of extracting temporal features from data sequences Xu et al. (2020). Particularly, they transform the observable data sequences into a latent feature space, where the featured distributions can be represented as a feature vector - including the temporal dimensional ones.

However, while this transformation effectively represents temporal dimensional features, the types of the significant features being extracted - whether static linearity or dynamic nonlinearity - depend on the model. Let's simplify RNNs in the form of $Y = f(\mathcal{X}; \theta)$, where the variable $\mathcal{X} = \langle X, t \rangle \in \mathbb{R}^{d+1}$ jointly represents observational-temporal features of the cause $X \in \mathbb{R}^d$. The optimization process of $\mathcal{X}$ is driven by the observational $Y$ through the relation $\theta$. Consequently, it can capture dynamical temporal features in the $t$-dimension if they are significant in predicting $Y$.

> **Remark 2.** RNNs extract dynamic nonlinearity from the cause by indexing via the relation $\theta$.

Despite their advantages, RNNs are not exempt from the *identifiability difficulty* Zhang (2012), primarily because of the requirement to specify timestamps for effects. This challenge primarily stems from the dynamical variations in the temporal dimension, brought by hidden $\omega$. Moreover, the difficulty intensifies when characterizing effects in comparison to causes. While it is feasible to organize sequential data around a major causal event (e.g., days of heavy rain), pinpointing the precise onset of subsequent effects (e.g., the exact day the flood initiated due to that rain) remains a complex task.

Given that the **relation-indexing** *autonomous identification* primarily targets dynamics of causes rather than effects, the inverse learning methodology Arora (2021), which has been gaining increasing attention recently, aspires to achieve the converse. It utilizes *symmetric directionality* to sidestep the challenge of identifying dynamical effects and defining the objective function.

> **Remark 3.** Autonomous indexing via objective relations $\theta$ can address the challenge of identifiability, but is not embraced by the current *Observation-Oriented* paradigm regarding *effects*.

Before the advent of RNNs, traditional methods typically utilize an observational time sequence to capture a set length of static features: Autoregressive models Hyvärinen (2010) are often formulated as $Y_{t+m} = f(\{x_t\}; \theta)$, while Granger causality Granger (1993); Maziarz (2015), a method widely recognized in economics, introduces another sequence for the effect, as $\{y_\tau\} = f(\{x_t\}; \theta)$, where $t$ and $\tau$ represent separate timelines for cause and effect. As highlighted in the discussions surrounding Figure 3, this method relies heavily on the precise specification regarding hidden relations $\omega$, and can hardly achieve generalizability autonomously.

To avoid specifying time sequences for causes, do-calculus Pearl (2012); Huang (2012) targets *identifiable* events, enabling a fluid transformation from dynamical cause to observational effect, but the identifiability relies on non-experimental data (controllable $\theta$). Given its inherently *differential* nature, which increases its complexity, we provide a streamlined reinterpretation of its three core rules from an *integral* viewpoint.

Let $do(x_t) = \{x_t, x_{t+1}\}$ indicate the occurrence of an instantaneous event $do(x)$ at time $t$, with the time step $\Delta t$ sufficiently small to make this event's *interventional* effect identifiable as a function of the resultant distribution at $t + 1$. Meanwhile, a separate *observational* effect is provoked by the static $x_t$. Then,

Given $\mathcal{X} \to Y \mid \theta$, where $\mathcal{X} = \langle X, t \rangle \in \mathbb{R}^{d+1}$ with augmented $t$-dimension residing a $T$-length sequence,

$$\mathcal{X} = \int_0^T do(x_t) \cdot x_t \, dt \quad \text{with} \quad \begin{cases} (do(x_t) = 1) \mid \theta, & \textit{Observational} \text{ only (Rule 1)} \\ (x_t = 1) \mid \theta, & \textit{Interventional} \text{ only (Rule 2)} \\ (do(x_t) = 0) \mid \theta, & \text{No } \textit{interventional} \text{ (Rule 3)} \\ \text{otherwise} & \text{Associated } \textit{observational} \text{ and } \textit{interventional} \end{cases}$$

The effect of $\mathcal{X}$ can be derived as $f(\mathcal{X}) = \int_0^T f_t\big(do(x_t) \cdot x_t\big) \, dt = \sum_{t=0}^{T-1} (y_{t+1} - y_t) = y_T - y_0$

Given a controllable $\theta$, it addresses three criteria that preserve conditional independence between *observational* and *interventional* effects, completing the chain rule, but sidesteps more generalized cases. If one depicts a dynamical effect as $\mathcal{Y} = \langle Y, \tau \rangle$, event specifications for $do(y)$ remain necessary.

### 3.3 Limitations in Application

Due to effect identification difficulties inherent within the *Observation-Oriented* paradigm, reliance on foundational assumptions is often indispensable. For a more detailed exploration of these limitations, Figure 5 categorizes the applications into four distinct scenarios: the queries can be divided into Discovery and Buildup, depending on whether the objective relation $\theta$ is known; they can also be further categorized by the dynamical significance of the effect - For example, the causal relationship "raining $\to$ wet floor" falls into area ④, while "raining $\to$ floor becoming wetter" is in area ③. They will be examined from two perspectives in the following: the modeling objective Relation (i.e., $\theta$), and the interpretational Directionality.

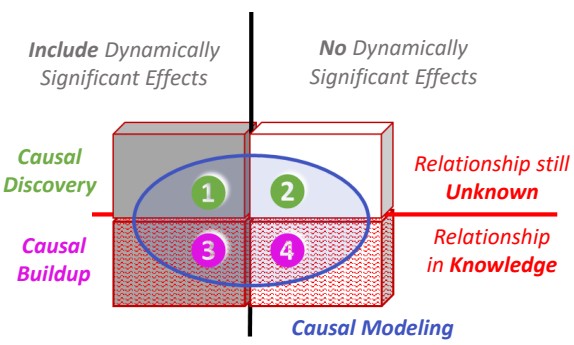

| | *Modeled Relation* | *Modeled Causal Direction* |
|---|---|---|
| **1** | Observational Only. Undiscovered Dynamics covered by *Faithfulness Assumption.* | Observational Data Determined. Not Logically Meaningful. |
| **2** | Observational Only. Aligned with Knowledge. | Observational Data Determined. Maybe Logically Suggestive. |
| **3** | Knowledge Determined. Unmodeled Dynamics covered by *Hidden Confounders* or *Sufficiency Assumption.* | Knowledge Determined. |
| **4** | Knowledge Determined. | Knowledge Determined. |

Figure 5: An overview of the current *Observation-Oriented* causal learning applications. The left rectangle cube represents all logical causal relationships, with the potentially modelable scope circled in blue.

*(1) Modeled Relation*

Knowledge primarily reflects a pre-determined causal model $f(;\theta)$ where the relation $\theta$ between observational entities $X$ and $Y$ is already established. While certain conditions allow for the transformation of some dynamics into observational forms - such as the independence in do-calculus - the model $f(;\theta)$ can still miss notable dynamics, particularly those in effects. Leveraging knowledge can enhance model interpretation by introducing hidden entities, such as $E$ depicted in Figure 4 (a). Without such interventions, these dynamics may be overlooked due to the causal *sufficiency* assumption, as seen with $\mathcal{B}_2$ in Figure 3 (b).

Data-driven causal discovery mainly investigates observational dependencies. When the true relationship of interest does not necessarily involve dynamical entities, the discovered associations (or correlations) can offer valuable insights. However, if such dynamics exist (especially within unobservable hierarchies), they may be dismissed by the causal *faithfulness* assumption, positing that observables can fully represent causal reality.

*(2) Modeled Causal Direction*

Consider observational entities $X$ and $Y$ with potential directional models $Y = f(X;\theta)$ and $X = g(Y;\phi)$, where $f(;\theta)$ and $g(;\phi)$ are pre-determined. The causal direction $X \to Y$ would be preferred if $\mathcal{L}(\hat{\theta}) > \mathcal{L}(\hat{\phi})$. Now, let $\mathcal{I}(\theta)$ be a simplified form of $\mathcal{I}_{X,Y}(\theta)$, the Fisher information representing $\theta$ given $\mathbf{P}(X,Y)$. If $p(\cdot)$ is the density function, then $\int_X p(x;\theta)dx$ is constant in this context. Thus, we have:

$$\mathcal{I}(\theta) = \mathbb{E}[(\frac{\partial}{\partial\theta}\log p(X,Y;\theta))^2 \mid \theta] = \int_Y \int_X (\frac{\partial}{\partial\theta}\log p(x,y;\theta))^2 p(x,y;\theta)dxdy$$

$$= \alpha \int_Y (\frac{\partial}{\partial\theta}\log p(y;x,\theta))^2 p(y;x,\theta)dy + \beta = \alpha\mathcal{I}_{Y|X}(\theta) + \beta, \text{with } \alpha, \beta \text{ constants.}$$

$$\text{Thus, } \hat{\theta} = \arg\max_\theta \mathbf{P}(Y \mid X, \theta) = \arg\min_\theta \mathcal{I}_{Y|X}(\theta) = \arg\min_\theta \mathcal{I}(\theta), \text{ and } \mathcal{L}(\hat{\theta}) \propto 1/\mathcal{I}(\hat{\theta}).$$

The inferred directionality depends on the extent to which the data informatively reflects the two opposing relations. Therefore, in purely data-driven causal discovery - where $\mathcal{I}(\theta) = \mathcal{I}(\phi) = 0$ contains no unobservable knowledge - the directionality is not logical but indicates the distributional dominance as determined by the data collection process, with the predominant one deemed the "cause". Even if $\theta$ and $\phi$ are knowledge-based, they remain only observationally meaningful, unnecessarily inferring true causal relations among dynamics.

## 4 The Overlooked Multi-Dimensional Temporal Space

As outlined in Definition 1, compared to our innate understanding of causal knowledge, an *Observation-Oriented* viewpoint has two key oversights: 1) the dynamical features of effects, and 2) the multi-dimensional nature of these dynamics. While the former still can be empirically addressed through inverse learning, the latter poses more foundational challenges to structural relationship modeling, underscoring the need for relation-indexing approaches in a new *Relation-Oriented* paradigm.

When understanding structural relationships within knowledge, our logic discerns not just the absolute timeline but also various relative timelines Coulson (2009), with each capturing distinct effects Shea (2001). While these effects may originate from a single cause, they possess unique dynamical features and interrelate with one another Wulf (1994). Within unobservable hierarchies, such interconnected dynamics (i.e., dynamically significant variables or features) can result in identical timestamps representing different effects across levels, inherently making the manual timestamp specification inaccurate.

Consider a structural causal relationship $\mathcal{Y} \xleftarrow{\theta} \mathcal{X} \xrightarrow{\phi} \mathcal{Z}$ comprising three dynamics $\{\mathcal{X}, \mathcal{Y}, \mathcal{Z}\}$ and two distinct effects on the relative timelines $T_\theta$ and $T_\phi$. Let's assume a hidden relation, $\omega$, introduces hierarchical levels. While it's feasible to model an individual effect on either $T_\theta$ or $T_\phi$ by specifying a sequence of timestamps, building a comprehensive structural model that encompasses $\{\mathcal{X}, \mathcal{Y}, \mathcal{Z}\}$ would introduce **inherent biases** if relying solely on any one timeline from either $T_\theta$ or $T_\phi$. Moreover, if $\langle \theta, \phi \rangle \in \mathbb{R}^h$ are jointly distributed, meaning $\mathcal{Y}$ and $\mathcal{Z}$ are interrelated, relying on a single timeline becomes unreliable even when considering individual effects. Only **autonomous identifications** specific to each effect can sidestep the complications.

Traditional causal inference, apparently aware of these challenges, has employed various de-confounding methods to circumvent these **confounded dynamics**, such as propensity score matching Benedetto (2018) and backdoor adjustment Pearl (2009). However, these techniques fundamentally depend on manual identification and have become impractical at present, given the black-box nature of AI models and their application to large-scale tasks. Consequently, while still operating under the *Observation-Oriented* paradigm, AI-based causal learning tends to default to the absolute timeline, which is the only directly observable one in the data, to specify timestamps for all events. This method can lead to *inherent biases* accumulating over the structural complexity, ultimately affecting the model's robustness and generalizability.

> **Definition 2.** The *Temporal Dimension* comprises all potential logical timelines, not a single dimension. A multi-dimensional Temporal Feature Space is defined by the required timelines serving as axes.

This section will first demonstrate the *inherent bias* through an intuitive example (subsection 4.1), explore its impact on the generalizability of structural causal models (subsection 4.2), and finally discuss the advancements and challenges on our path toward causal knowledge-aligned AI (subsection 4.3).

### 4.1 Scheme of the Inherent Bias

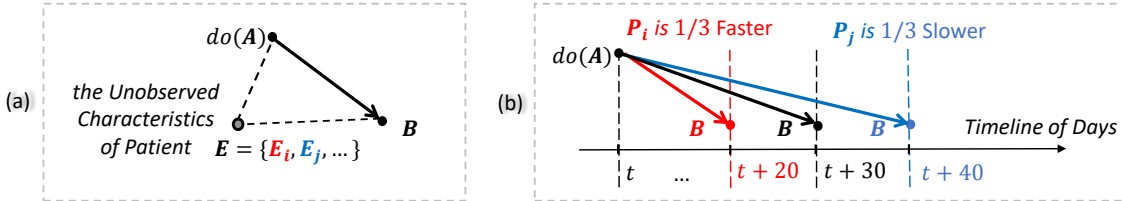

Figure 6: (a) Initial DAG introducing hidden $E$. (b) Enhanced DAG (Directed Acyclic Graph).

Consider medical trial data from hospital patients. Vital signs and medication usage are recorded daily, forming the chronological *absolute timeline.* However, to assess the effects of a specific medication, a *relative timeline* is constructed, with time-zero marking a consistent action, such as $do(A)$, for all patients. As a result, events with different chronological timestamps can align on the relative timeline, and vice versa. For instance, Figure 3 illustrates a relative timeline for the effects of $do(A)$, while Figure 6(a) revisits its causal

DAG, incorporating the introduced hidden confounder. For clearly represent hierarchical dynamical effects, the causal DAG is enhanced as depicted in (b) through two steps:

1. Assume dynamically significant effects and integrate their relative timelines into the DAG space.
2. Use varying edge lengths to represent timespans required for the effects to reach an equivalent magnitude.

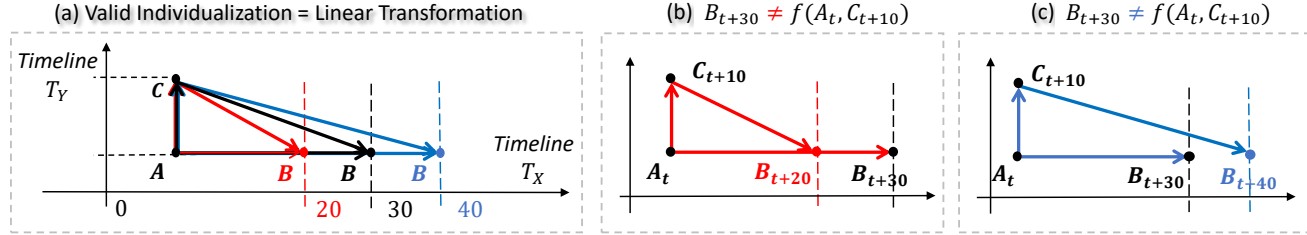

Figure 7: (a) A two-timeline DAG space with valid individualization processes. (b) and (c) Violations of the *Markov* condition in the SCM when specifying timestamps under the *Observation-Oriented* paradigm.

Figure 7(a) presents an extended scenario where $A$ stands for $do(A)$ for short. It features two distinct effects: the primary effect $\overrightarrow{AB}$ on $B$, and a side effect $\overrightarrow{AC}$ on vital sign $C$, which indirectly affects $B$ through $\overrightarrow{CB}$. The confounded nodes $\{A, B, C\}$ form a triangle across timelines $T_X$ and $T_Y$, which should consistently hold for all individuals or populations, based on the causal *Markov* condition requirement. The processes of *generalization* and *individualization* operate as "stretching" this triangle along $T_X$ at different ratios, conducting a homographic *linear transformation* within this DAG space, as depicted in Figure 7 (a).

For simplicity, assume dynamics on $T_X$ and $T_Y$ are independent: fix the $\overrightarrow{AC}$ timespan at 10 days for all patients, focusing on individualized variances only on $T_X$. Structural Causal Models (SCMs) typically assign a timespan for $\overrightarrow{AB}$, such as 30, to represent the population-level average effect. However, as shown in (b) and (c), the SCM function $B_{t+30} = f(A_t, C_{t+10})$ violates the *Markov* condition for either $P_i$ or $P_j$.

> **Remark 4.** The ***inherent bias*** may occur in SCM if it contains: 1) *Confounded Dynamics* across *Multiple* logical timelines, and 2) Unobservable Hierarchy (represented by hidden $\omega$).

In this simplified scenario, SCMs might still function given the independence between $\overrightarrow{AB}$ and $\overrightarrow{AC}$. However, it is impractical always to assume independence or a lack of confounding for all dynamical effects in structural relationship learning. For broad causal AI applications, neglecting multiple relative timelines can lead to accumulating biases, potentially compromising model robustness irrespective of the chosen model. Consequently, current AI applications typically focus on tasks not involving relative timelines. For instance, LLMs operate within a semantic space on a single timeline, consistently preserving word order.

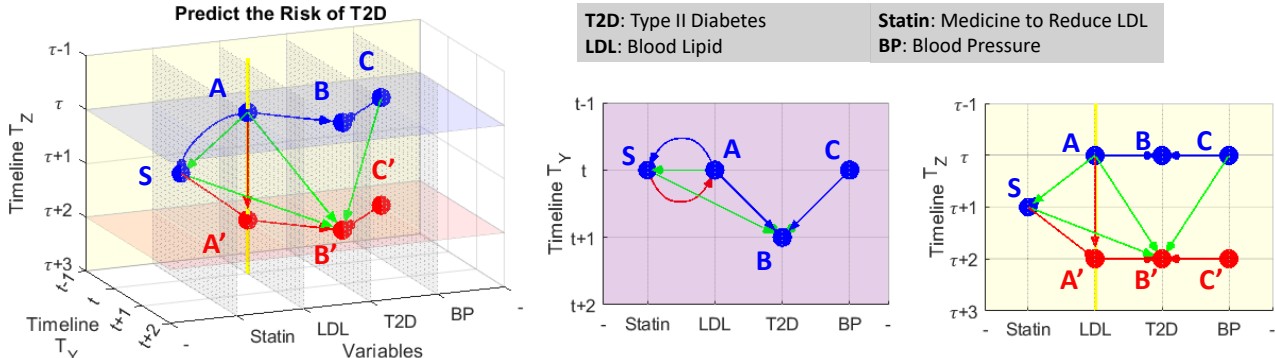

Figure 8: A 3D temporal DAG space with two timelines $\mathcal{T}_Y$ and $\mathcal{T}_Z$. The specified SCM $B' = f(A, C, S)$ evaluates Statin's medical effect on reducing T2D risk. On $\mathcal{T}_Y$, the step $\Delta t$ from $t$ to $(t+1)$ allows $A$ and $C$ to fully influence $B$. The step $\Delta \tau$ on $\mathcal{T}_Z$, from $(\tau+1)$ to $(\tau+2)$, let Statin fully release to forward $A$ to $A'$.

## 4.2 Inherent Impact on SCM Generalizability

Unobservable hierarchies can imply varied scenarios with the same fundamental relationships. Traditional SCMs, which necessitate timestamp specification along a singular $t$-timeline, compromise not only the robustness but also impede the generalizability of the formulated SCMs across these scenarios.

Consider the practical scenario depicted in Figure 8. Here, $\Delta t$ and $\Delta \tau$ represent actual time spans. Yet, the crux is not on determining their exact values, but on realizing their intended causal relationship: As each unit of Statin's effect is delivered on LDL via $\overrightarrow{SA'}$, it immediately impacts T2D through $\overrightarrow{A'B'}$. Simultaneously, the next unit effect begins generation. This dual action runs concurrently until $S$ is fully administered. At $B'$, the ultimate aim of this process is to evaluate the total cumulative influence stemming from $S$.

Given the relationship $\overrightarrow{SB'} = \overrightarrow{SA'} + \overrightarrow{A'B'}$, specifying the $\overrightarrow{SB'}$ time span (= half of the $\overrightarrow{AB'}$ time span) inherently sets the $\Delta t : \Delta \tau$ ratio, defining the $ASB'$ triangle's shape in the DAG space. While the estimated mean effect at $B'$ might be precise for the present population, the preset $\Delta t : \Delta \tau$ ratio's universality is questionable, potentially constraining the established SCM's generalizability.

## 4.3 Toward Causal Knowledge-Aligned AI

In pursuit of causally interpretable AI, our modeling techniques expand beyond the purely observational to encompass temporal dimensions, as summarized in Figure 9. At present, the challenge is to ensure the generalizability of structural causal AI models. Recognizing multi-timeline dynamics is essential to avoid biases that obscure AI interpretability. Given the impracticality of manually discerning all potential logical timelines for observable data, it might be time to contemplate a new paradigm.

| Model | Principle | Cause | Relation & Direction | Effect | Handle Unobservable Hierarchy | Capture Dynamics |
|---|---|---|---|---|---|---|
| Mechanistic or Physical | $\mathcal{Y} = f(\mathcal{X}; \theta)$ | Observational-Temporal $\mathcal{X} = \langle X, t \rangle$ | by Knowledge | Observational-Temporal $\mathcal{Y} = \langle Y, t \rangle$ | Yes | Yes |
| Relation-Indexing Methodology | Given $\boldsymbol{P}(\mathcal{X}, \mathcal{Y})$ & $\mathcal{X} \to \mathcal{Y}$ | Observational-Temporal $\mathcal{X} = \langle X, t \rangle$ | by Representation $\hat{\mathcal{Y}} = f(\mathcal{X}; \theta)$ | Observational-Temporal $\hat{\mathcal{Y}} = \langle \hat{Y}, t \rangle$ | Yes | Yes |
| Structural Causal Learning | Given $\boldsymbol{P}(X, Y)$ & $X \to Y$ $Y = f(X; \theta)$ | Observational Sequence $\{X_t\}$ | $X \to Y$ via Relation $\theta$ by Knowledge | Observational and Static $Y_t$ | ? | ? |
| Graphical Causal Discovery | Given $\boldsymbol{P}(X, Y)$ Find $\mathcal{L}(Y\|X; \theta) > \mathcal{L}(X\|Y; \theta)$ | Observational $X$ | Observationally Associated $X$ and $Y$ | Observational $Y$ | ? | No |
| Common Cause Model | Given $\boldsymbol{P}(X, Y\|Z)$ | Observational $X$ | Related via $Z$ | Observational $Y$ | ? | No |
| i.i.d Data Driven Model | Given $\boldsymbol{P}(X, Y)$ | Observational $X$ | None | Observational $Y$ | No | No |

Figure 9: Simple taxonomy of models (adapted in part of Table 1 in Scholkopf (2021)), from more knowledge-driven (top in purple) to more data-driven (bottom in green). Notations: $\theta$ = parameter derived from joint or conditional distribution, $\langle X, t \rangle$ = augment $t$-dimension, "?" = depending on practice.

The initial models under i.i.d. assumption only approximate observational associations, proved unreliable for causal reasoning Pearl et al. (2000); Peters et al. (2017). Correspondingly, the common cause principle highlights the significance of the nontrivial conditional properties, to distinguish structural relationships from statistical dependencies Dawid (1979); Geiger & Pearl (1993), providing a basis for effectively uncovering the underlying structures in graphical models Peters et al. (2014).

Graphical causal models relying on conditional dependencies to construct Bayesian networks (BNs) often operate in observational space and neglect temporal aspects, reducing their causal relevance Scheines (1997). Causally significant models, such as Structural Equation Models (SEMs) and Functional Causal Models (FCMs) Glymour et al. (2019); Elwert (2013), can address counterfactual queries Scholkopf (2021), with respect to temporal distributions by leveraging prior knowledge, to construct causal DAGs accordingly.

State-of-the-art deep learning applications on causality, which encode the DAG structural constraint into continuous optimization functions Zheng et al. (2018; 2020); Lachapelle et al. (2019), undoubtedly enable highly efficient solutions, especially for large-scale problems. However, larger question scales indicate more underlying logical timelines, which may lead to snowballing temporal biases. It can be evident from the limited successful applications of incorporating DAG structure into network architectures Luo (2020); Ma (2018), e.g., neural architecture search (NAS).

Schölkopf Scholkopf (2021) summarized three key challenges impeding causal AI applications to achieving generalizable success: 1) limited model robustness, 2) insufficient model reusability, and 3) inability to handle data heterogeneity (caused by unobservable hierarchies in knowledge). There exists an intrinsic connection between these challenges and the inherent bias highlighted in Remark 4.

On the other side, physical models, which explicitly integrate temporal dimensions in computation, and are able to establish abstract concepts through relations, may provide insights into these challenges. The relation-indexing approach is designed to bridge the gap between the Observational and Temporal Spaces.

## Chapter II: Realization of Proposed Relation-Oriented Paradigm

This chapter delves into the realization of autonomously identifying causal effects via relation-indexing, and its role in shaping structural models in the latent feature space. First, Section 5 details the technique for extracting relation-indexed representations, to realize hierarchical disentanglement. Building on this, Section 6 presents the *Relation-Indexed Representation Learning* (RIRL) method, designed to instantiate structural causal models within latent space. Lastly, Section 7 provides experimental validation of RIRL's efficacy.

## 5 Relation-Indexed Hierarchical Disentanglement

In the relationship $\mathcal{X} \to \mathcal{Y}$, we define dynamics $\mathcal{X} = \langle X, t \rangle \in \mathbb{R}^{d+1}$ and $\mathcal{Y} = \langle Y, \tau \rangle \in \mathbb{R}^{b+1}$, given observational variables, $X \in \mathbb{R}^d$ and $Y \in \mathbb{R}^b$, respectively. For $\mathcal{X}$, the data is stored as the time series $\{x_t\} = \{x_1, \ldots, x_{T_x}\}$ with a length of $T_x$, which can also be viewed as a vector $\overrightarrow{x}$ of dimensionality $d * T_x$ in the *observable data space*. Similarly, $\mathcal{Y}$ is stored as the data sequence $\{y_\tau\} = \{y_1, \ldots, y_{T_y}\}$ with a length of $T_y$, and can be observed as a $(b * T_y)$-dimensional vector $\overrightarrow{y}$. Notably, $t$ and $\tau$ are two separate timelines.

Relation-indexing begins with **_initializations_** of $\mathcal{X}$ and $\mathcal{Y}$ as the latent space features, $\mathcal{H} \in \mathbb{R}^L$ and $\mathcal{V} \in \mathbb{R}^L$, respectively. A relation model denoted as $f(; \theta)$, then refines $\mathcal{H}$ and $\mathcal{V}$ to minimize their distance in $\mathbb{R}^L$. So, the dimensionality $L$ of the *latent feature space* $\mathbb{R}^L$ must be at least the rank of the augmented triplet, given by $L \geq rank(\langle \mathcal{X}, \theta, \mathcal{Y} \rangle)$, raising a technical challenge to represent $\overrightarrow{x}$ and $\overrightarrow{y}$ in higher-dimensional features.

> **Remark 5.** The variable *initialization* necessitates a *higher-dimensional* representation autoencoder.

The goal of relation-indexing is to obtain $\hat{\mathcal{Y}}$, which is the component of $\mathcal{Y}$ that can be **_identifiable_** through its relationship with $\mathcal{X}$, accordingly represented as $\hat{\mathcal{V}}$ in the latent feature space. Moreover, for the relationship models to be *generalizable*, $\hat{\mathcal{V}}$ must serve as a basis, which permits subsequent components of $\mathcal{Y}$ to build upon it, representing its various other relationships, leading to the *hierarchical disentanglement* of $\mathcal{Y}$.

### 5.1 Higher-Dimensional Autoencoder

Autoencoders are commonly used for dimensionality reduction, especially in applications involving multiple observables formulating structural models Wang (2016). Our *Relation-Oriented* approach, in contrast, aims to sequentially model individual relationships within a higher-dimensional space $\mathbb{R}^L$, and simultaneously "stack" them to construct the structure within $\mathbb{R}^L$.

Figure 10 illustrates the autoencoder architecture designed for achieving this higher-dimensional representation. This architecture is featured by the symmetrical *Expander* and *Reducer* layers (source code is available [1]). The Expander magnifies the input vector $\overrightarrow{x}$ by capturing its higher-order associative features, while the

---

[1] https://github.com/kflijia/bijective_crossing_functions/blob/main/code_bicross_extracter.py

Reducer symmetrically diminishes dimensionality and reverts to its initial state. For precise reconstruction, the ***invertibility*** of these processes is essential.

The Expander showcased in Figure 10 implements a *double-wise* expansion. Here, every duo of digits from $\overrightarrow{x}$ is encoded into a new digit using an association with a random constant, termed the *Key*. This *Key* is generated by the encoder and replicated by the decoder. Such pairwise processing of $\overrightarrow{x}$ expands its length from $(d*T_x)$ to be $(d*T_x-1)^2$. By leveraging multiple *Keys* and concatenating their resultant vectors, $\overrightarrow{x}$ can be considerably expanded, ready for the subsequent dimensionality-reduced representation extraction. The four blue squares with unique grid patterns represent expansions by four distinct *Keys*, with the grid patterns acting as their "signatures". Each square symbolizes a $(d*T_x-1)^2$ length vector. Similarly, higher-order expansions, like *triple-wise* across three digits, can be achieved with adapted *Keys*.

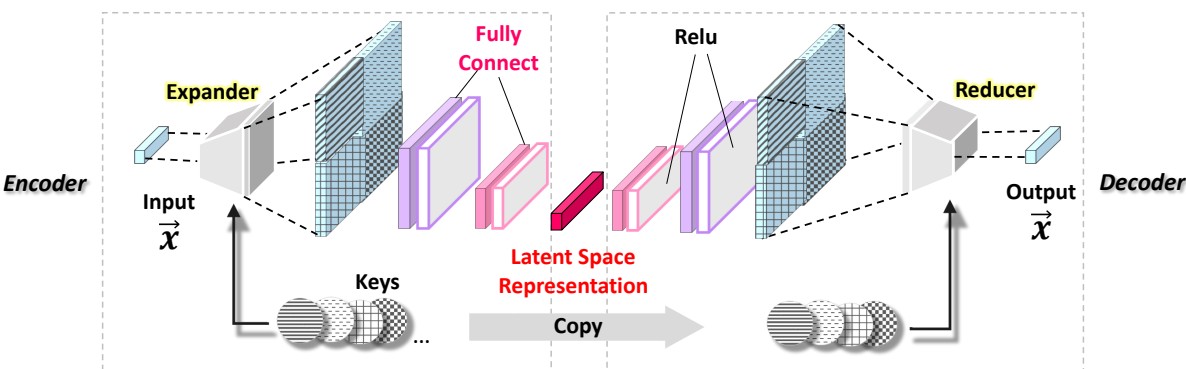

Figure 10: *Invertible* autoencoder architecture for extracting *higher-dimensional* representations.

Figure 11 illustrates the encoding and decoding processes within the Expander and Reducer, targeting the digit pair $(x_i, x_j)$ for $i \neq j \in 1, \ldots, d$. The Expander function is defined as $f_\theta(x_i, x_j) = x_j \otimes exp(s(x_i)) + t(x_i)$, which hinges on two elementary functions, $s(\cdot)$ and $t(\cdot)$. The *Key* parameter, $\theta$, embodies their weights, $\theta = (w_s, w_t)$. Specifically, the Expander morphs $x_j$ into a new digit $y_j$ utilizing $x_i$ as a chosen attribute. In contrast, the Reducer symmetrically uses the inverse function $f_\theta^{-1}$, defined as $(y_j - t(y_i)) \otimes exp(-s(y_i))$. This method avoids calculating $s^{-1}$ or $t^{-1}$, granting flexibility for nonlinear transformations to $s(\cdot)$ and $t(\cdot)$. This design is inspired by the pioneering work of Dinh et al. (2016) on invertible neural network layers that utilize bijective functions.

## 5.2 Relation-Indexed Representation

Consider $x$ and $y$ as the instances of $\mathcal{X}$ and $\mathcal{Y}$, respectively, with their corresponding vector representations $h$ and $v$ in $\mathbb{R}^L$. The latent dependency $\mathbf{P}(v|h)$ is utilized for training the relation function $f_\theta = f(; \theta)$, as illustrated in Figure 12. During each iteration, the learning process undergoes three optimization steps:

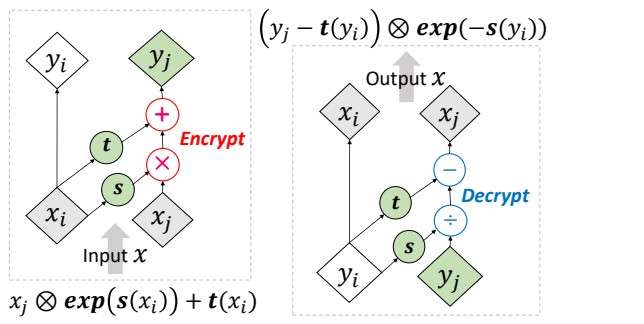

Figure 11: Expander (left) and Reducer (right).

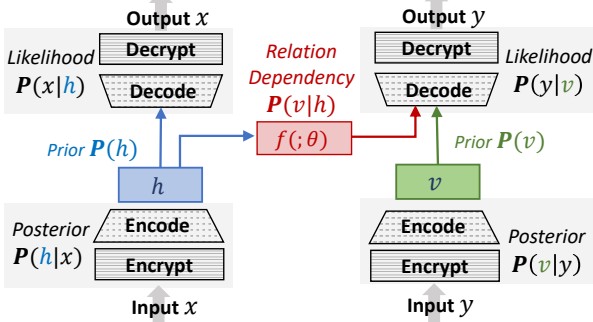

Figure 12: Relationship model architecture.

1. Optimizing the cause-encoder by $\mathbf{P}(h|x)$, the relation model by $\mathbf{P}(v|h)$, and the effect-decoder by $\mathbf{P}(y|v)$ to reconstruct the relationship $x \to y$, represented as $h \to v$ in $\mathbb{R}^L$.
2. Fine-tuning the effect-encoder $\mathbf{P}(v|y)$ and effect-decoder $\mathbf{P}(y|v)$ to accurately represent $y$.
3. Fine-tuning the cause-encoder $\mathbf{P}(h|x)$ and cause-decoder $\mathbf{P}(x|h)$ to accurately represent $x$.

During the learning process, the values of $h$ and $v$ are iteratively adjusted to reduce their distance in $\mathbb{R}^L$. The relation function $f_\theta = f(;\theta)$ serves as a bridge to span this distance. It effectively represents the hyper-dimensional variable $\theta \in \mathbb{R}^h$ as an index, guiding the output of $f_\theta$ to encapsulate the associated representation $\langle \hat{\mathcal{H}}, \theta, \hat{\mathcal{V}} \rangle$. From $\hat{\mathcal{V}}$, the effect component $\hat{\mathcal{Y}}$ can be reconstructed. Within the system, for each effect, a series of such relation functions $\{f_\theta\}$ is maintained, indexing diverse levels of causal inputs for sequentially building the structural model.

## 5.3 Hierarchical Disentanglement of Effects

Consider $\mathcal{Y} = \langle Y, t \rangle \in \mathbb{R}^{b+1}$ having an $n$-level hierarchy, with each level built using a representation function, labeled as $g(;\omega_i)$ for the $i$-th level. For clarity, just use $\omega_i$ to represent the $i$-th level feature in the *latent feature space* $\mathbb{R}^L$; its counterpart in the *observable data space* $\mathbb{R}^{b+1}$ is denoted as $\Omega_i$ (i.e., $\hat{\mathcal{Y}}$ at the $i$-th level).

Let the vector $\omega_i$ in $\mathbb{R}^L$ primarily spans a sub-dimensional space, $\mathbb{R}^{L_i}$. This results in the hierarchical disentanglement sequence $\{\mathbb{R}^{L_1}, \ldots, \mathbb{R}^{L_i}, \ldots, \mathbb{R}^{L_n}\}$ that fully represent $\mathcal{Y}$. Function $g_i$ maps from $\mathbb{R}^{b+1}$ to $\mathbb{R}^{L_i}$, taking into account features from all previous levels as attributes. This gives us:

$$\mathcal{Y} = \sum_{i=1}^n \Omega_i, \text{ where } \Omega_i = g_i\big(\omega_i;\ \Omega_1, \ldots, \Omega_{i-1}\big) \text{ with } \Omega_i \in \mathbb{R}^{d+1} \text{ and } \omega_i \in \mathbb{R}^{L_i} \subseteq \mathbb{R}^L \tag{1}$$

The $i$-th component in the *observable data space*, denoted as $\Omega_i \in \mathbb{R}^{d+1}$, is articulated through an observational data sequence with the length of $T_y$, along the absolute timeline $t$. However, in latent space, the objective of $\omega_i$ is to capture dynamics along a relative timeline, $t_i$, which is autonomously determined by the relation at the $i$-th level, not bound by the observational timestamps in $\mathbb{R}^{d+1}$.

In the context of a purely observational hierarchy, with $\mathcal{Y}$ substituted by $Y \in \mathbb{R}^b$, Figure 2 (b) can be interpreted as follows: Consider three feature levels represented as $\omega_1 \in \mathbb{R}^{L_1}$, $\omega_2 \in \mathbb{R}^{L_2}$, and $\omega_3 \in \mathbb{R}^{L_3}$. For simplicity, assume each subspace is mutually exclusive, so that $L = L_1 + L_2 + L_3$. In the latent space, the triplet $\langle \omega_1, \omega_2, \omega_3 \rangle \in \mathbb{R}^L$ comprehensively depicts the image. Their observable counterparts, $\Omega_1$, $\Omega_2$, and $\Omega_3$, are three distinct full-scale images, each showcasing different content. For example, $\Omega_1$ emphasizes finger details, while the combination $\Omega_1 + \Omega_2$ reveals the entire hand.

# 6 RIRL: Building Structural Models in Latent Space

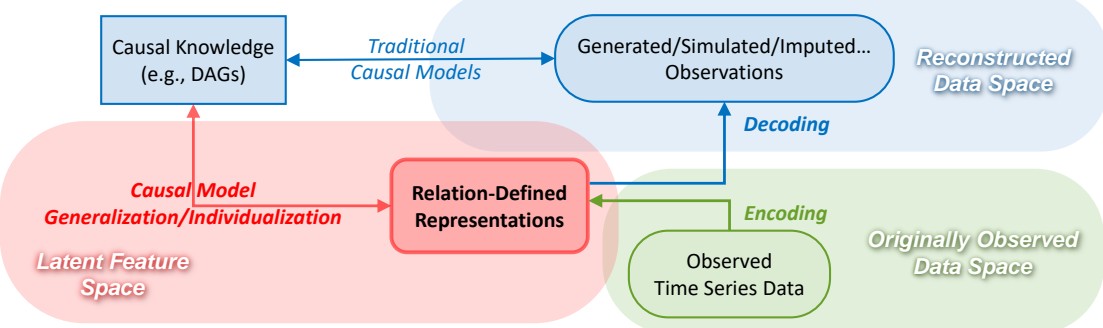

Figure 13: How Relation-Indexed Representation Learning (RIRL) contributes to traditional models.

By sequentially stacking relation-indexed representations, causal structural models can be established as aligned with causal knowledge. Figure 13 illustrates how the RIRL method seals the black-box nature of AI within the latent space, while simultaneously generating interpretable observations that enhance existing

*Observation-Oriented* models, such as conducting on-demand counterfactual simulations. These cryptic representations, though opaque to humans, can internally promote model generalization and individualization, managed exclusively within the AI's latent space.

This Section first presents the method to construct structural relationship models in the latent space (subsection 6.1), and describes the technique for discovering structures within the latent space by identifying potential relationships among initialized variable representations (subsection 6.2).

## 6.1 Stacking Hierarchical Representations

A structural relationship can be represented by a causal graph, denoted as $G$. To construct models in the latent space, the latent dimensionality $L$ must be sufficiently large to adequately represent $G$. Let's denote a data matrix augmented by all observational attributes in $G$ as $\mathbf{X}$. Given the need to include informative relations $\{\theta\}$ for the edges in $G$, it is essential that $L > rank(\mathbf{X})+1$, where the $+1$ accounts for the $t$-timeline.

The PCA principle posits that the space $\mathbb{R}^L$ learned by the autoencoder is spanned by the top principal components of $\mathbf{X}$ Baldi (1989); Plaut (2018); Wang (2016). Hypothetically, reducing $L$ below $rank(\mathbf{X})$ may yield a less adequate but causally more significant latent space through better alignment of dimensions Jain (2021) (Further exploration in this direction is warranted). Bypassing a deep dive into dimensionality boundaries, we rely on empirical fine-tuning for the experiments in this study (reducing $L$ from 64 to 16).

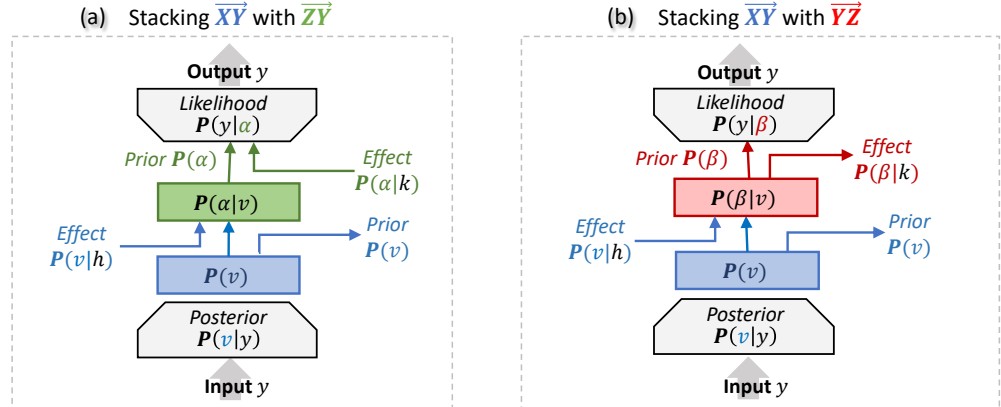

Figure 14: Stacking relation-indexed representations to construct hierarchy.

Consider the structural causal relationship among dynamically significant variables $\{\mathcal{X}, \mathcal{Y}, \mathcal{Z}\}$, each having corresponding representations $\{\mathcal{H}, \mathcal{V}, \mathcal{K}\} \in \mathbb{R}^L$ initially derived from three autoencoders. Figure 14 illustrates the hierarchical assembly of two modeled relationships associated with $\mathcal{Y}$.

In Figure 14, two stacking scenarios are displayed based on varying causal directions. With the established $\mathcal{X} \to \mathcal{Y}$ relationship in $\mathbb{R}^L$, the left-side architecture finalizes the $\mathcal{X} \to \mathcal{Y} \leftarrow \mathcal{Z}$ structure, while the right-side focuses on $\mathcal{X} \to \mathcal{Y} \to \mathcal{Z}$. Through the addition of a representation layer, hierarchical disentanglement is formed, allowing for various input-output combinations (denoted as $\mapsto$) according to specific requirements.

For example, on the left, $\mathbf{P}(v|h) \mapsto \mathbf{P}(\alpha)$ represents the $\mathcal{X} \to \mathcal{Y}$ relationship, whereas $\mathbf{P}(\alpha|k)$ implies $\mathcal{Z} \to \mathcal{Y}$. Conversely, on the right, $\mathbf{P}(v) \mapsto P(\beta|k)$ denotes the $\mathcal{Y} \to \mathcal{Z}$ relationship with $\mathcal{Y}$ as input. Meanwhile, $\mathbf{P}(v|h) \mapsto P(\beta|k)$ captures the causal sequence $\mathcal{X} \to \mathcal{Y} \to \mathcal{Z}$.

## 6.2 Causal Discovery in Latent Space

Algorithm 1 outlines the heuristic procedure for identifying edges among the initial variable representations. We use Kullback-Leibler Divergence (KLD) as a metric to evaluate the strength of causal relationships. Specifically, as depicted in Figure 12, KLD evaluates the similarity between the RNN output $\mathbf{P}(v|h)$ and the prior $\mathbf{P}(v)$. Lower KLD values indicate stronger causal relationships due to closer alignment with the ground truth. Conversely, while Mean Squared Error (MSE) is a frequently used evaluation metric, its sensitivity

to data variances Reisach (2021); Kaiser & Sipos (2021) leads us to utilize it as a supplementary measure in this study.

---

**Algorithm 1:** Latent Space Causal Discovery

---

**Result:** ordered edges set $\mathbf{E} = \{e_1, \ldots, e_n\}$
$\mathbf{E} = \{\}$ ; $N_R = \{n_0 \mid n_0 \in N, Parent(n_0) = \varnothing\}$ ;
**while** $N_R \subset N$ **do**
    $\Delta = \{\}$ ;
    **for** $n \in N$ **do**
        **for** $p \in Parent(n)$ **do**
            **if** $n \notin N_R$ $and$ $p \in N_R$ **then**
                $e = (p, n)$; $\beta = \{\}$;
                **for** $r \in N_R$ **do**
                    **if** $r \in Parent(n)$ $and$ $r \neq p$ **then**
                        $\beta = \beta \cup r$
                  **end**
                **end**
                $\delta_e = K(\beta \cup p, n) - K(\beta, n)$;
                $\Delta = \Delta \cup \delta_e$;
            **end**
        **end**
    **end**
    $\sigma = argmin_e(\delta_e \mid \delta_e \in \Delta)$;
    $\mathbf{E} = \mathbf{E} \cup \sigma$; $N_R = N_R \cup n_\sigma$;
**end**

| | |
|---|---|
| $G = (N, E)$ | graph $G$ consists of $N$ and $E$ |
| $N$ | the set of nodes |
| $E$ | the set of edges |
| $N_R$ | the set of reachable nodes |
| $\mathbf{E}$ | the list of discovered edges |
| $K(\beta, n)$ | KLD metric of effect $\beta \to n$ |
| $\beta$ | the cause nodes |
| $n$ | the effect node |
| $\delta_e$ | KLD Gain of candidate edge $e$ |
| $\Delta = \{\delta_e\}$ | the set $\{\delta_e\}$ for $e$ |
| $n,p,r$ | notations of nodes |
| $e,\sigma$ | notations of edges |

Figure 15 illustrates the causal structure discovery process in latent space over four steps. Two edges, ($e_1$ and $e_3$), are sequentially selected, with $e_1$ setting node $B$ as the starting point for $e_3$. In step 3, edge $e_2$ from $A$ to $C$ is deselected and reassessed due to the new edge $e_3$ altering $C$'s existing causal conditions. The final DAG represents the resulting causal structure.

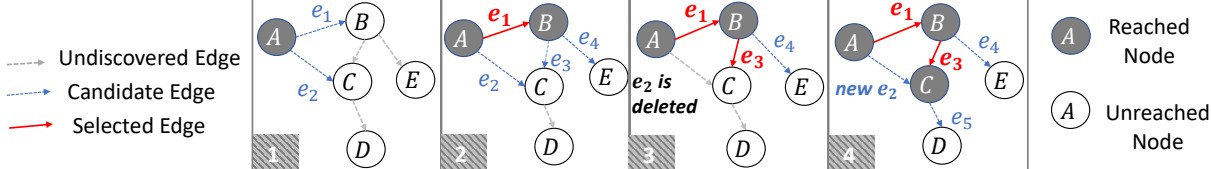

Figure 15: An example of causal discovery in the latent space.

## 7 Efficacy Validation Experiments

The experiments aim to validate the efficacy of the RIRL method from three aspects: 1) the performance of the proposed higher-dimensional representations, evaluated by reconstruction accuracy, 2) the construction of a clear effect hierarchy through the stacking of relation-indexed representations, and 3) the identification of DAG structures within the latent space through discovery. A full demonstration of the conducted experiments is available online [2]. The experiments in this study present two primary limitations, detailed as follows:

Firstly, the dataset used in the current experiments may not be optimal for assessing the efficacy of RIRL. In particular, real-world causal data, such as clinical records, often contain inherent biases. While empirical constraints limited our access to such data for this study, the synthetic data we utilized may not be ideal for validating the improved model robustness conferred by RIRL. For experiments that validate the presence of such inherent biases, readers are referred to prior research Li et al. (2020).

Secondly, the time windows designated for cause and effect, $T_x$ and $T_y$, are consistently set at 10 and 1, respectively. This constraint arose from an initial oversight in the experimental design, wherein the pivotal role of *dynamics* was not fully recognized, leading to restrictions set by the RNN pattern. This limitation manifests when constructing causal sequences, such as in $\mathcal{X} \to \mathcal{Y} \to \mathcal{Z}$. While the model adeptly captures single-hop effects, it struggles with two-hop information due to the dynamics in $\mathcal{Y}$ being segmented into

---

[2]https://github.com/kflijia/bijective_crossing_functions.git

statics by the effect window $T_y = 1$, resulting in a loss of dynamic information. However, extending the length of $T_y$ does not pose a significant technical challenge to future works.

### 7.1 Hydrology Dataset

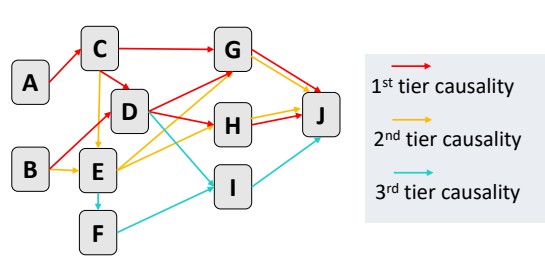

| ID | Variable Name | Explanation |
|----|---------------|-------------|
| A | Environmental set I | Wind Speed, Humidity, Temperature |
| B | Environmental set II | Temperature, Solar Radiation, Precipitation |
| C | Evapotranspiration | Evaporation and transpiration |
| D | Snowpack | The winter frozen water in the ice form |
| E | Soil Water | Soil moisture in vadose zone |
| F | Aquifer | Groundwater storage |
| G | Surface Runoff | Flowing water over the land surface |
| H | Lateral | Vadose zone flow |
| I | Baseflow | Groundwater discharge |
| J | Streamflow | Sensors recorded outputs |

Figure 16: Hydrological causal DAG: routine tiers organized by descending causal strength.

The dataset chosen for our experiments is a widely-used synthetic resource in the field of hydrology, aimed at enhancing streamflow predictions based on observed environmental conditions such as temperature and precipitation. In hydrology, deep learning, particularly RNN models, has gained favor for extracting observational representations and predicting streamflow Goodwell (2020); Kratzert (2018). We focus on a simulation of the Root River Headwater watershed in Southeast Minnesota, covering 60 consecutive virtual years with daily updates. The simulated data is from the Soil and Water Assessment Tool (SWAT), a comprehensive system grounded in physical modules, to generate dynamically significant hydrological time series.

Figure 16 displays the causal DAG employed by SWAT, complete with node descriptions. The hydrological routines are color-coded based on their contribution to output streamflow. Surface runoff (1st tier) significantly impacts rapid streamflow peaks, followed by lateral flow (2nd tier). Baseflow dynamics (3rd tier) have a subtler influence. Our causal discovery experiments aim to reveal these underlying tiers.

### 7.2 Higher-Dimensional Variable Representation Test

In this test, we have a total of ten variables (or nodes), each requiring a separate autoencoder for initializing a higher-dimensional representation. Table 1 lists the statistics of their post-scaled (i.e., normalized) attributes, as well as their autoencoders' reconstruction accuracies. Accuracy is assessed in the root mean square error (RMSE), where a lower RMSE indicates higher accuracy for both scaled and unscaled data.

The task is challenging due to the limited dimensionalities of the ten variables - maxing out at just 5 and the target node, $J$, having just one attribute. To mitigate this, we duplicate the input vector to a consistent 12-length and add 12 dummy variables for months, resulting in a 24-dimensional input. A double-wise extension amplifies this to 576 dimensions, from which a 16-dimensional representation is extracted via the autoencoder. Another issue is the presence of meaningful zero-values, such as node $D$ (Snowpack in winter), which contributes numerous zeros in other seasons and is closely linked to node $E$ (Soil Water). We tackle this by adding non-zero indicator variables, called *masks*, evaluated via binary cross-entropy (BCE).

Despite challenges, RMSE values ranging from 0.01 to 0.09 indicate success, except for node $F$ (the Aquifer). Given that aquifer research is still emerging (i.e., the 3rd tier baseflow routine), it is likely that node $F$ in this synthetic dataset may better represent noise than meaningful data.

### 7.3 Hierarchical Disentanglement Test

Table 3 provides the performance comparison of stacking relation-indexed representations on each node. The term "single-effect" is to describe the accuracy of a specific effect node when reconstructed from a single cause node (e.g., $B \rightarrow D$ and $C \rightarrow D$), and "full-effect" for the accuracy when all its cause nodes are stacked

Table 1: Statistics of variable attributes and performances of the variable representation test.

| Variable | Dim | Mean | Std | Min | Max | Non-Zero Rate% | RMSE on Scaled | RMSE on Unscaled | BCE of Mask |
|---|---|---|---|---|---|---|---|---|---|
| A | 5 | 1.8513 | 1.5496 | -3.3557 | 7.6809 | 87.54 | 0.093 | 0.871 | 0.095 |
| B | 4 | 0.7687 | 1.1353 | -3.3557 | 5.9710 | 64.52 | 0.076 | 0.678 | 1.132 |
| C | 2 | 1.0342 | 1.0025 | 0.0 | 6.2145 | 94.42 | 0.037 | 0.089 | 0.428 |
| D | 3 | 0.0458 | 0.2005 | 0.0 | 5.2434 | 11.40 | 0.015 | 0.679 | 0.445 |
| E | 2 | 3.1449 | 1.0000 | 0.0285 | 5.0916 | 100 | 0.058 | 3.343 | 0.643 |
| F | 4 | 0.3922 | 0.8962 | 0.0 | 8.6122 | 59.08 | 0.326 | 7.178 | 2.045 |
| G | 4 | 0.7180 | 1.1064 | 0.0 | 8.2551 | 47.87 | 0.045 | 0.81 | 1.327 |
| H | 4 | 0.7344 | 1.0193 | 0.0 | 7.6350 | 49.93 | 0.045 | 0.009 | 1.345 |
| I | 3 | 0.1432 | 0.6137 | 0.0 | 8.3880 | 21.66 | 0.035 | 0.009 | 1.672 |
| J | 1 | 0.0410 | 0.2000 | 0.0 | 7.8903 | 21.75 | 0.007 | 0.098 | 1.088 |

Table 2: Brief summary of the latent space causal discovery test.

| Edge | A→C | B→D | C→D | C→G | D→G | G→J | D→H | H→J | B→E | E→G | E→H | C→E | E→F | F→I | I→J | D→I |
|---|---|---|---|---|---|---|---|---|---|---|---|---|---|---|---|---|
| KLD | 7.63 | 8.51 | 10.14 | 11.60 | 27.87 | 5.29 | 25.19 | 15.93 | 37.07 | 39.13 | 39.88 | 46.58 | 53.68 | 45.64 | 17.41 | 75.57 |
| Gain | 7.63 | 8.51 | 1.135 | 11.60 | 2.454 | 5.29 | 25.19 | 0.209 | 37.07 | -5.91 | -3.29 | 2.677 | 53.68 | 45.64 | 0.028 | 3.384 |

(e.g., $BC \to D$). To provide context, we also include baseline performance scores based on the initial variable representations. During the relation learning process, the effect node serves two purposes: it maintains its own accurate representation (as per optimization no.2 in 5.2) and helps reconstruct the relationship (as per optimization no.1). Both aspects are evaluated in Table 3.

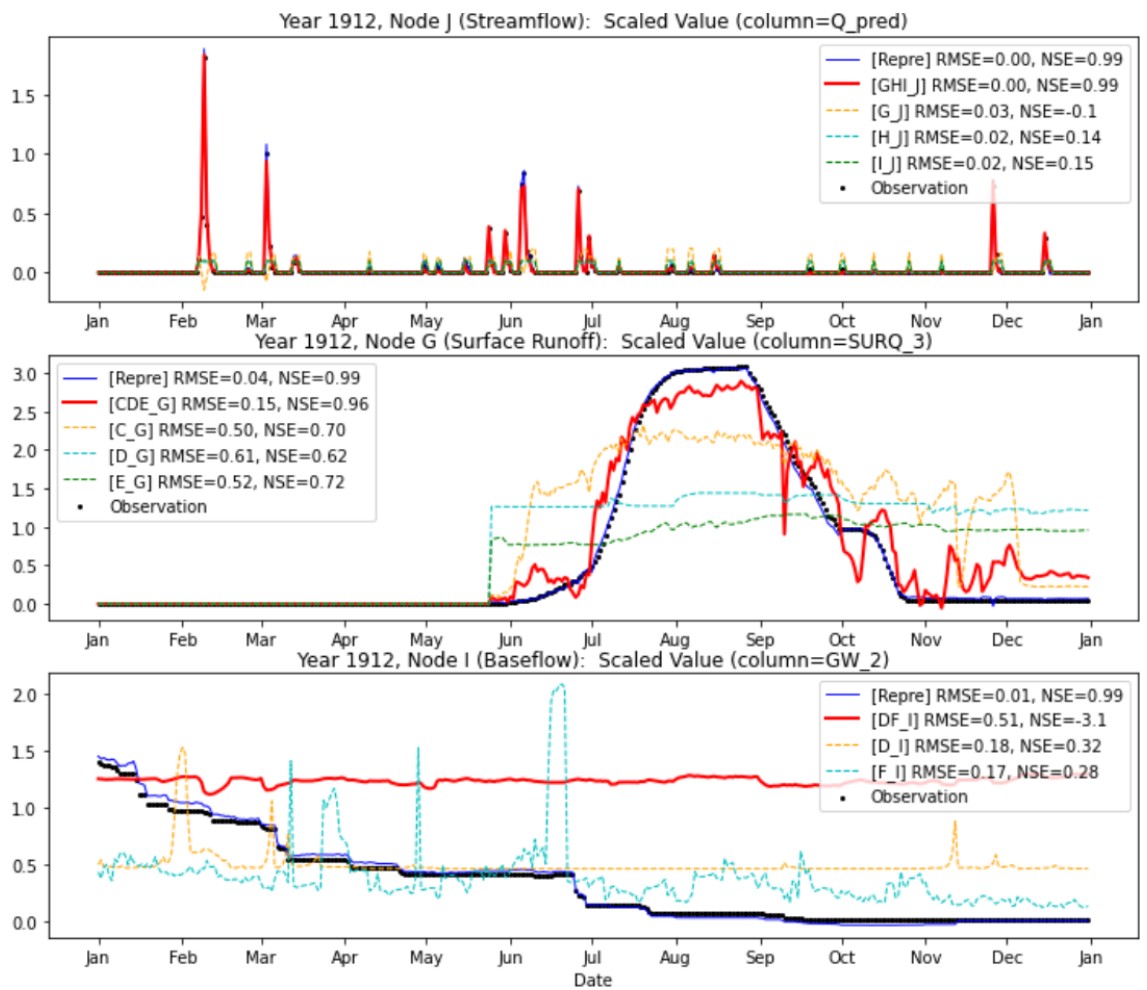

Figure 17: Reconstructed dynamical effects, via hierarchically stacked relation-indexed representations.

Table 3: Effect Reconstruction Performances of RIRL sorted by effect nodes.

| Result Node | Variable Representation (Initial) | | | Cause Node | Variable Representation (in Relation Learning) | | | Relationship Reconstruction | | | |
|---|---|---|---|---|---|---|---|---|---|---|---|
| | RMSE | | BCE | | RMSE | | BCE | RMSE | | BCE | KLD |
| | on Scaled Values | on Unscaled Values | Mask | | on Scaled Values | on Unscaled Values | Mask | on Scaled Values | on Unscaled Values | Mask | (in latent space) |
| C | 0.037 | 0.089 | 0.428 | A | 0.0295 | 0.0616 | 0.4278 | 0.1747 | 0.3334 | 0.4278 | 7.6353 |
| D | 0.015 | 0.679 | 0.445 | BC | 0.0350 | 1.0179 | 0.1355 | 0.0509 | 1.7059 | 0.1285 | 9.6502 |
| | | | | B | 0.0341 | 1.0361 | 0.1693 | 0.0516 | 1.7737 | 0.1925 | 8.5147 |
| | | | | C | 0.0331 | 0.9818 | 0.3404 | 0.0512 | 1.7265 | 0.3667 | 10.149 |
| E | 0.058 | 3.343 | 0.643 | BC | 0.4612 | 26.605 | 0.6427 | 0.7827 | 45.149 | 0.6427 | 39.750 |
| | | | | B | 0.6428 | 37.076 | 0.6427 | 0.8209 | 47.353 | 0.6427 | 37.072 |
| | | | | C | 0.5212 | 30.065 | 1.2854 | 0.7939 | 45.791 | 1.2854 | 46.587 |
| F | 0.326 | 7.178 | 2.045 | E | 0.4334 | 8.3807 | 3.0895 | 0.4509 | 5.9553 | 3.0895 | 53.680 |
| G | 0.045 | 0.81 | 1.327 | CDE | 0.0538 | 0.9598 | 0.0878 | 0.1719 | 3.5736 | 0.1340 | 8.1360 |
| | | | | C | 0.1057 | 1.4219 | 0.1078 | 0.2996 | 4.6278 | 0.1362 | 11.601 |
| | | | | D | 0.1773 | 3.6083 | 0.1842 | 0.4112 | 8.0841 | 0.2228 | 27.879 |
| | | | | E | 0.1949 | 4.7124 | 0.1482 | 0.5564 | 10.852 | 0.1877 | 39.133 |
| H | 0.045 | 0.009 | 1.345 | DE | 0.0889 | 0.0099 | 2.5980 | 0.3564 | 0.0096 | 2.5980 | 21.905 |
| | | | | D | 0.0878 | 0.0104 | 0.0911 | 0.4301 | 0.0095 | 0.0911 | 25.198 |
| | | | | E | 0.1162 | 0.0105 | 0.1482 | 0.5168 | 0.0097 | 3.8514 | 39.886 |
| I | 0.035 | 0.009 | 1.672 | DF | 0.0600 | 0.0103 | 3.4493 | 0.1158 | 0.0099 | 3.4493 | 49.033 |
| | | | | D | 0.1212 | 0.0108 | 3.0048 | 0.2073 | 0.0108 | 3.0048 | 75.577 |
| | | | | F | 0.0540 | 0.0102 | 3.4493 | 0.0948 | 0.0098 | 3.4493 | 45.648 |
| J | 0.007 | 0.098 | 1.088 | GHI | 0.0052 | 0.0742 | 0.2593 | 0.0090 | 0.1269 | 0.2937 | 5.5300 |
| | | | | G | 0.0077 | 0.1085 | 0.4009 | 0.0099 | 0.1390 | 0.4375 | 5.2924 |
| | | | | H | 0.0159 | 0.2239 | 0.4584 | 0.0393 | 0.5520 | 0.4938 | 15.930 |
| | | | | I | 0.0308 | 0.4328 | 0.3818 | 0.0397 | 0.5564 | 0.3954 | 17.410 |

The KLD metrics in Table 3 indicate the strength of learned causality, with a lower value signifying stronger. For instance, node $J$'s minimal KLD values suggest a significant effect caused by nodes $G$ (Surface Runoff), $H$ (Lateral), and $I$ (Baseflow). In contrast, the high KLD values imply that predicting variable $I$ using $D$ and $F$ is challenging. For nodes $D$, $E$, and $J$, the "full-effect" are moderate compared to their "single-effect" scores, suggesting a lack of informative associations among the cause nodes. In contrast, for nodes $G$ and $H$, lower "full-effect" KLD values imply capturing meaningful associative effects through hierarchical stacking. The KLD metric also reveals the most contributive cause node to the effect node. For example, the proximity of the $C \rightarrow G$ strength to $CDE \rightarrow G$ suggests that $C$ is the primary contributor to this causal relationship.

Figure 17 showcases reconstructed time series, for the effect nodes $J$, $G$, and $I$, in the same synthetic year to provide a straightforward overview of the hierarchical representation performances. Here, black dots represent the ground truth; the blue line indicates reconstruction via the initial variable representation, and the "full-effect" representation generates the red line. In addition to RMSE, we also employ the Nash–Sutcliffe model efficiency coefficient (NSE) as an accuracy metric, commonly used in hydrological predictions. The NSE ranges from $-\infty$ to 1, with values closer to 1 indicating higher accuracy.

The initial variable representation closely aligns with the ground truth, as shown in Figure 17, attesting to the efficacy of our proposed autoencoder architecture. As expected, the "full-effect" performs better than the "single-effect" for each effect node. Node $J$ exhibits the best prediction, whereas node $I$ presents a challenge. For node $G$, causality from $C$ proves to be significantly stronger than the other two, $D$ and $E$.

## 7.4 Latent Space Causal Discovery Test

The discovery test initiates with source nodes $A$ and $B$ and proceeds to identify potential edges, culminating in the target node $J$. Candidate edges are selected based on their contributions to the overall KLD sum (less gain is better). Table 6 shows the order in which existing edges are discovered, along with the corresponding KLD sums and gains after each edge is included. Color-coding in the cells corresponds to Figure 16, indicating tiers of causal routines. The arrangement underscores the efficacy of this latent space discovery approach.

A comprehensive list of candidate edges evaluated in each discovery round is provided in Table 4 in Appendix A. For comparative purposes, we also performed a 10-fold cross-validation using the conventional FGES discovery method; those results are available in Table 5 in Appendix A.

## 8   Conclusions

The concept of Artificial General Intelligence (AGI) has sparked extensive discussions over the years Marcus (2020). Recent debates have particularly focused on whether large language models (LLMs) edge us closer to realizing AGI Schaeffer et al. (2023). A central question is whether symbols, as well as symbol-grounded systems, such as AI, can represent our empirical understanding and inquiries Newell (2007); Pavlick (2023).

We posit that the core challenge is representing the "human understanding" process symbolically, particularly in symbolizing those abstract, intangible concepts within our cognition. Specifically, we require a framework that can intuitively formalize our essential appeals underlying learning inquiries.

This study introduces the concept of "hyper-dimension" to symbolize the abstract information (termed as "relations") present in our cognition, representing them as distributional variables, $\langle \theta, \omega \rangle \in \mathbb{R}^h$. This allows them to interface directly with the conventional variables we are familiar with in modeling, which reside in the "observational-temporal dimensions", integrally denoted as $\mathcal{Y} = f(\mathcal{X}; \theta)$ with $\langle \theta, \omega \rangle \in \mathbb{R}^h$.

The proposed *dimensionality framework* seeks to unify a range of learning inquiries, from traditional causal inference to modern AI Alignment challenges. In doing so, it underscores two pivotal factors that the current relationship learning paradigm frequently overlooks: the "dynamics" and the "relative timelines" they span. Classical statistics recognize the intertwined timelines, focusing on manually identifying cut-off points (i.e., de-confounding), but lack the capability to handle inherent dynamics (i.e., temporal non-linearities). On the other hand, mainstream AI approaches primarily concentrate on exploring non-linear associations in the observational dimensions, often simplifying multiple potential timelines into a singular one. For certain applications, such as LLMs, employing a singular timeline is suitable, enabling AI to discern meaningful associations that shed light on the latent causal knowledge. However, while these associations over $\{\mathcal{X}, \mathcal{Y}\}$ might align with the *unobservable knowledge* represented by $\langle \theta, \omega \rangle \in \mathbb{R}^h$, they do not truly encapsulate it, leading to a perception that AI can generate intelligent responses without truly understanding the content.

While there have been attempts to address these issues (for instance, the introduction of hierarchical temporal memory in neuroscience Wu (2018)), we assert that human logic discerns timelines via relations. These relations not only give shape to our logical structures but also infuse our models with the essence of knowledge alignment. The journey to achieving AGI will undoubtedly be a historically extensive and complex undertaking, necessitating a vast array of knowledge-aligned AI model constructions. This study aspires to establish foundational insights for future developments in the field.

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

# A   Appendix: Complete Experimental Results of Causal Discovery

Table 4: The Complete Results of Heuristic Causal Discovery in latent space. Each row stands for a round of detection, with '#" identifying the round number, and all candidate edges are listed with their KLD gains as below. 1) Green cells: the newly detected edges. 2) Red cells: the selected edge. 3) Blue cells: the trimmed edges accordingly.

| | | | | | | | | | | | | | | | | |
|---|---|---|---|---|---|---|---|---|---|---|---|---|---|---|---|---|
| **# 1** | **A→ C** | A→ D | A→ E | A→ F | **B→ C** | B→ D | B→ E | B→ F | | | | | | | | |
| | **7.6354** | 19.7407 | 60.1876 | 119.7730 | **8.4753** | 8.5147 | 65.9335 | 132.7717 | | | | | | | | |
| **# 2** | A→ D | A→ E | A→ F | B→ D | B→ E | B→ F | C→ D | C→ E | C→ F | C→ G | C→ H | C→ I | | | | |
| | 19.7407 | 60.1876 | 119.7730 | **8.5147** | 65.9335 | 132.7717 | 10.1490 | 46.5876 | 111.2978 | 11.6012 | 39.2361 | 95.1564 | | | | |
| **# 3** | **A→ D** | A→ E | A→ F | B→ E | B→ F | C→ D | C→ E | C→ F | C→ G | C→ H | C→ I | D→ E | D→ F | D→ G | D→ H | D→ I |
| | **9.7357** | 60.1876 | 119.7730 | 65.9335 | 132.7717 | **1.1355** | 46.5876 | 111.2978 | 11.6012 | 39.2361 | 95.1564 | 63.7348 | 123.3203 | 27.8798 | 25.1988 | 75.5775 |
| **# 4** | A→ E | A→ F | B→ E | B→ F | C→ E | C→ F | **C→ G** | C→ H | C→ I | D→ E | D→ F | D→ G | D→ H | D→ I | | |
| | 60.1876 | 119.7730 | 65.9335 | 132.7717 | 46.5876 | 111.2978 | **11.6012** | 39.2361 | 95.1564 | 63.7348 | 123.3203 | 27.8798 | 25.1988 | 75.5775 | | |
| **# 5** | A→ E | A→ F | B→ E | B→ F | C→ E | C→ F | C→ H | C→ I | D→ E | D→ F | D→ G | D→ H | D→ I | G→ J | | |
| | 60.1876 | 119.7730 | 65.9335 | 132.7717 | 46.5876 | 111.2978 | 39.2361 | 95.1564 | 63.7348 | 123.3203 | **2.4540** | 25.1988 | 75.5775 | 5.2924 | | |
| **# 6** | A→ E | A→ F | B→ E | B→ F | C→ E | C→ F | C→ H | C→ I | D→ E | D→ F | D→ H | D→ I | G→ J | | | |
| | 60.1876 | 119.7730 | 65.9335 | 132.7717 | 46.5876 | 111.2978 | 39.2361 | 95.1564 | 63.7348 | 123.3203 | 25.1988 | 75.5775 | **5.2924** | | | |
| **# 7** | A→ E | A→ F | B→ E | B→ F | C→ E | C→ F | **C→ H** | C→ I | D→ E | D→ F | D→ H | D→ I | | | | |
| | 60.1876 | 119.7730 | 65.9335 | 132.7717 | 46.5876 | 111.2978 | **39.2361** | 95.1564 | 63.7348 | 123.3203 | **25.1988** | 75.5775 | | | | |
| **# 8** | A→ E | A→ F | B→ E | B→ F | C→ E | C→ F | C→ I | D→ E | D→ F | D→ I | H→ J | | | | | |
| | 60.1876 | 119.7730 | 65.9335 | 132.7717 | 46.5876 | 111.2978 | 95.1564 | 63.7348 | 123.3203 | 75.5775 | **0.2092** | | | | | |
| **# 9** | **A→ E** | A→ F | B→ E | B→ F | C→ E | C→ F | C→ I | D→ E | D→ F | D→ I | | | | | | |
| | **60.1876** | 119.7730 | 65.9335 | 132.7717 | **46.5876** | 111.2978 | 95.1564 | 63.7348 | 123.3203 | 75.5775 | | | | | | |
| **# 10** | A→ F | B→ E | B→ F | C→ F | C→ I | **D→ E** | D→ F | D→ I | E→ F | E→ G | E→ H | E→ I | | | | |
| | 119.7730 | **-6.8372** | 132.7717 | 111.2978 | 95.1564 | **17.0407** | 123.3203 | 75.5775 | 53.6806 | -5.9191 | -3.2931 | 110.2558 | | | | |
| **# 11** | A→ F | B→ F | C→ F | C→ I | D→ F | D→ I | E→ F | **E→ G** | E→ H | E→ I | | | | | | |
| | 119.7730 | 132.7717 | 111.2978 | 95.1564 | 123.3203 | 75.5775 | 53.6806 | **-5.9191** | -3.2931 | 110.2558 | | | | | | |
| **# 12** | A→ F | B→ F | C→ F | C→ I | D→ F | D→ I | E→ F | **E→ H** | E→ I | | | | | | | |
| | 119.7730 | 132.7717 | 111.2978 | 95.1564 | 123.3203 | 75.5775 | 53.6806 | **-3.2931** | 110.2558 | | | | | | | |
| **# 13** | **A→ F** | **B→ F** | **C→ F** | C→ I | **D→ F** | D→ I | **E→ F** | E→ I | | | | | | | | |
| | **119.7730** | **132.7717** | **111.2978** | 95.1564 | **123.3203** | 75.5775 | **53.6806** | 110.2558 | | | | | | | | |
| **# 14** | C→ I | D→ I | **E→ I** | **F→ I** | | | | | | | | | | | | |
| | 95.1564 | 75.5775 | **110.2558** | **45.6490** | | | | | | | | | | | | |
| **# 15** | C→ I | D→ I | **I→ J** | | | | | | | | | | | | | |
| | 15.0222 | 3.3845 | **0.0284** | | | | | | | | | | | | | |
| **# 16** | **C→ I** | **D→ I** | | | | | | | | | | | | | | |
| | **15.0222** | **3.3845** | | | | | | | | | | | | | | |

Table 5: Average performance of 10-Fold FGES (Fast Greedy Equivalence Search) causal discovery, with the prior knowledge that each node can only cause the other nodes with the same or greater depth with it. An edge means connecting two attributes from two different nodes, respectively. Thus, the number of possible edges between two nodes is the multiplication of the numbers of their attributes, i.e., the lengths of their data vectors. (All experiments are performed with 6 different Independent-Test kernels, including chi-square-test, d-sep-test, prob-test, disc-bic-test, fisher-z-test, myplr-test. But their results turn out to be identical.)

| Cause Node | A | B | | C | | | D | | | E | | | F | G | H | I |
|---|---|---|---|---|---|---|---|---|---|---|---|---|---|---|---|---|
| True Causation | A→C | B→D | B→E | C→D | C→E | C→G | D→G | D→H | D→I | E→F | E→G | E→H | F→I | G→J | H→J | I→J |
| Number of Edges | 16 | 24 | 16 | 6 | 4 | 8 | 12 | 12 | 9 | 8 | 8 | 8 | 12 | 4 | 4 | 3 |
| Probability of Missing | 0.038889 | 0.125 | 0.125 | 0.062 | 0.06875 | 0.039286 | 0.069048 | 0.2 | 0.142857 | 0.3 | 0.003571 | 0.2 | 0.142857 | 0.0 | 0.072727 | 0.030303 |
| Wrong Causation | | | | C→F | | | | D→E | D→F | | | F→G | G→H | G→I | H→I | |
| Times of Wrongly Discovered | | | | 5.6 | | | | 1.2 | 0.8 | | | 5.0 | 8.2 | 3.0 | 2.8 | |

Table 6: Brief Results of the Heuristic Causal Discovery in latent space, identical with Table 3 in the paper body, for better comparison to the traditional FGES methods results on this page. The edges are arranged in detected order (from left to right) and their measured causal strengths in each step are shown below correspondingly. Causal strength is measured by KLD values (less is stronger). Each round of detection is pursuing the least KLD gain globally. All evaluations are in 4-Fold validation average values. Different colors represent the ground truth causality strength tiers (referred to the Figure 10 in the paper body).

| Causation | A→C | B→D | C→D | C→G | D→G | G→J | D→H | H→J | C→E | B→E | E→G | E→H | E→F | F→I | I→J | D→I |
|---|---|---|---|---|---|---|---|---|---|---|---|---|---|---|---|---|
| KLD | 7.63 | 8.51 | 10.14 | 11.60 | 27.87 | 5.29 | 25.19 | 15.93 | 46.58 | 65.93 | 39.13 | 39.88 | 53.68 | 45.64 | 17.41 | 75.57 |
| Gain | 7.63 | 8.51 | 1.135 | 11.60 | 2.454 | 5.29 | 25.19 | 0.209 | 46.58 | -6.84 | -5.91 | -3.29 | 53.68 | 45.64 | 0.028 | 3.384 |

