# OpenReview forum: "Relation-Oriented: Toward Causal Knowledge-Aligned AGI"
_TMLR — Rejected by TMLR_

### Review · Reviewer_js92 · 2023-08-16

**Summary Of Contributions:**

In this paper, authors argue that relation-oriented perspective is more informative than observation-oriented principle. Then, they proposed relation-oriented modeling framework and design relation-defined representation learning. Experiments on synthetic dataset are conducted.

**Audience:**

Yes

**Claims And Evidence:**

No

**Requested Changes:**

Please see weaknesses.

**Strengths And Weaknesses:**

Strengths
1)	The topic “toward knowledge-aligned causal AI” is fundamental to AI and seems to be urgent for GPT models.
2)	Authors think deeply on a series of topics related to AI risks, such as AI misalignment, division of cognitive space, relation-defined representation, knowledge hierarchy, etc. . Their statements are insightful and might be helpful for innovations in future.
3)	The presentation is good. Several nice figures are provided. It is easy to follow.

Weaknesses
1)	Key concepts and claims are not rigorously defined, presented, or proved. For example, what is the definition of “temporal bias” and “relation”? Thus, I cannot understand what lemma 1-3 claims in logical. And there is no proof. It is hard to check whether they are reasonable.
2)	The logic flow is not clear among section 3-5. Why these discussions are necessary for the main proposal “relation-oriented learning”? Some of the arguments seems not sound. For example, for SCM, we can also model timeline into it like chapter 10 in the book “Elements of causal inferences: foundations and learning algorithms”. For the hierarchical factorization formula in sec 5, it seems that the timeline/hyper-dimension is not essential. Can we take the knowledge in different levels as unobservable variables in SCM with in the increasing causal order?
3)	What are main differences of the proposed framework in figure 12? Can we construct keys automatically for general AI tasks?
4)	In section 6.3, is there any identifiability guarantee for the causal discovery in latent space? Does the proposed method also encounter the identifiability difficulty of causal representation learning?
5)	No experiments on real AI tasks.

To summarize, the motivation is strong and the topic is important. But the main proposal lacks of rigorous statement, which makes it hard to check its soundness. The novelty and technical contribution of the new algorithms seems to be limited, especially without the empirical justifications on real AI tasks.

---

> ### Author Response · Authors · 2023-08-17
>
> $\textbf{Q1}$: “for SCM, we can also model timeline into it like chapter 10 in the book ‘Elements of causal inferences' ”
>
> $\textbf{A1}$: Yes, it is exactly the “single absolute timeline” challenged through this paper (try searching for “Granger”). Unless you can provide a solution for Figure 8's violations using existing models (including but not limited to those in your mentioned chapter 10).
>
> $\\ $
>
> $\textbf{Q2}$: “Key concepts … are not rigorously defined, …, what is the definition of “temporal bias”?”
>
> $\textbf{A2}$: If you have suggestions for naming the bias in Figure 8, caused by the violations, I'm open to hearing them. Otherwise, please prove the violations can be avoided using existing methods.
>
> $\\ $
>
> $\textbf{Q3}$: “The logic flow is not clear among section 3-5. Why these discussions are necessary for the main proposal ‘relation-oriented learning’?”
>
> $\textbf{A3}$:
>
> $\{Section\ 3}$: to state that the current “causal learning” and “causal discovery” are indeed “correlation learning” and “informativeness discovery”, utilizing assumptions to eliminate causal-relative aspects and then pretending they are solving “causal” – If you suggest me to speak it out, please let me know.
>
> $\{Section\ 4}$: to argue that the “single absolute timeline” you proposed can lead to unsolvable violations with all existing methods, unless you can resolve the example in Figure 8.
>
> $\{Section\ 5}$: to show how to factorize the (n+1) dimensional <X,t> in latent space, rather than the n-dimensional X, as no one has previously done this. Though it may seem insignificant, its absence can become significant for reviewers.
>
> $\\ $
>
> $\textbf{Q4}$: “Does the proposed method also encounter the identifiability difficulty of causal representation learning?”
>
> $\textbf{A4}$: Indexing (i.e., identifying) through relations is the only way to solve this difficulty – The content in $\{Section\ 3-5}$.
>
> $\\ $
>
> $\textbf{Q5}$: “Can we take the knowledge in different levels as unobservable variables in SCM with in the increasing causal order?”
>
> $\textbf{A5}$: Sure, if in this way you can avoid the Figure 8 example’s violations.
>
> $\\ $
>
> $\textbf{Q6}$: “without the empirical justifications on real AI tasks”
>
> $\textbf{A6}$: Calling for AI tasks is the purpose of this paper, as stated in conclusion, and underscored by the deficiencies noted in the experiments.
>
> It’s like, affirming Heliocentric needs someone to initially speak out “Earth is not the center”, rather than waiting a decade for Copernicus to complete an evidence book.
>
> In the realm of AI, we've only tapped into 2 rich mines (language & image) out of a potential 100, with 98 hidden. This paper presents a possible key to unlock those hidden areas, as to prompt others to notice, validate, and refine this key, towards the hidden 98, rather than letting redundant efforts within the known 2 to contribute to the creation of flawed knowledge, while crafting a perfect key myself.
> To counter the notion of these hidden opportunities, please address the example in Figure 8.
>
> $\\ $
>
>
> $\textbf{Q7}$: “what is the definition of “relation”?”
>
> $\textbf{A7}$: For a philosophical definition: See the first citation on Page 3.
>
> For an empirical definition: See the cited book “Elements of causal inferences”, also the one you mentioned.
>
> For a simple definition: Use your common sense (which will also aid in understanding Lemmas 1-3). For example, if you have a son named James, your relation with him defines him as "son" to you, rather than any James. If you need more examples, please let me know.
>
> $\\ $
>
>
> $\textbf{Q8}$:…
>
> How about you attempt to solve the example in Figure 8 first? I appreciate the challenges you've presented, and I'm eager to engage in further discussion. Thank you!

---

> ### Author Response · Authors · 2023-08-17
>
> Current models predominantly follow the paradigm: $Y_{t+m}=f(X_t)$ with a specified $t$-value as an attribute. Instead of focusing on the traditional question, “How can we improve $f$ for a specific $t$?” I urge you to consider a deeper problem.
>
> As illustrated on Page 8, the moment we need to specify $t$, we become like ants on a tree named $t$, and the violations depicted in Figure 8 become unavoidable.
>
> The solution is not to perfect the function, but to remove the need to $t$-value altogether, treating it as a computational dimension. How can we do this? By identifying $t$-distributions through relations. Why? This reflects how human knowledge is constructed through relationships, as discussed in the philosophical paper cited on Page 3.
>
> If you know of any models that exist outside of this paradigm, I would be grateful if you could share them with me.
>
> This paper does not follow the typical structure of “Here's an important question! -> I propose a new method! -> See! It’s better than the others!” Instead, reading it requires only common sense and basic comprehension of English. For instance, terms like "identifiable difficulty" are simply other ways to express the problem – provided, of course, that you read the material thoroughly.
>
> I genuinely hope that future challenges will be more along the lines of "Your reasoning is flawed here!" rather than "This doesn't align with the textbook!" If, however, your journal only accepts papers that follow a conventional pattern, please do inform me. Thank you!

---

> ### Author Response · Authors · 2023-09-06
>
> $\textbf{Q8}$: “What are main differences of the proposed framework in figure 12? Can we construct keys automatically for general AI tasks?”
>
> $\textbf{A8}$: My apologies for any confusion. The framework is outlined in Figure 11, while Figure 12 presents the proposed architecture.
>
> For Figure 11: The proposed framework enables AI systems to generate counterfactual effects on-demand, something that existing AI models struggle with. This is evident when we see fields such as health informatics still sticking with traditional statistical methods for counterfactuals, despite the surge of AI in accuracy-focused applications.
>
> For Figure 12: Keys are randomized values and can be designed as per the specific requirements of any AI task, so what do you mean by “automatically”? In terms of differences, consider this: Can a standard autoencoder derive 64-dimensional representations from 16-dimensional data?
>
> $\\ $
>
> $\textbf{Q9}$: “In section 6.3, is there any identifiability guarantee for the causal discovery in latent space?”
>
> $\textbf{A9}$:
> First, what makes the identifiability difficult in causal discovery? – The reasons are detailed in Sections 3 and 4; Once you delve, the necessity for operating in the latent space should become evident.
>
> $\\ $
>
> $\textbf{Q10}$: Further philosophical explanation for the “relation” concept.
>
> $\textbf{A10}$: I designed the paper primarily for readers rooted in computer science, intentionally veering away from deeper philosophical tangents. However, if you believe it's essential to delve deeper, I'll consider including the content below.
>
> Within the modeling context, “relations” pertain to the unobservable facets of our knowledge. Consider the example, “you have a son named James”. To AI, the unique bond between a parent and child is intangible. Of course, AI may deduce a special connection between you two from observed interactions, but this doesn't equate to discerning the true “parent-child” relationship.
> The current modeling approaches often involve humans to manually identify interactions at first, for teaching AI the “parent-child” relationship. This paper proposes the opposite: Provide AI with the “parent-child” relation as an index, enabling it to identify these interactions autonomously.
>
> While this may not alter the learning results in this straightforward example, when dealing with causality involving temporal dimensions, the differences become stark. Only the relation-indexed method proposed can yield accurate outcomes, whereas the conventional methodology is riddled with biases, exemplified in Figure 8.
>
> The notion of “multiple levels of knowledge” is also a type of “relation”, depicting a cause-effect hierarchy, just as your mentioned “increasing causal order”. However, in the modeling context, our target is to generalize across these levels rather than modeling their inner connection. Therefore, aiming for more accessible comprehension, I labeled it as a “hierarchy” in the paper.
> Per Figure 1, the “hyper-dimensional space” symbolizes the collective of all unobservable “relations” within our knowledge, though I've kept this definition implicit to maintain clarity.
>
> Do share any feedback or further inquiries. I appreciate your insights!

---

### Review · Reviewer_jFmP · 2023-09-08

**Summary Of Contributions:**

The paper aims to introduce a "Relation-Oriented" perspective to machine learning, challenging the traditional "Observation-Oriented" models. The authors propose a new method for representation learning that they claim can better capture complex relationships and temporal dynamics. However, the paper's ambitious scope and language may come across as overreaching, and a thorough evaluation of the proposed methodology is needed to substantiate the claims made.

The paper's main contributions are:

1. Introduction of a "Relation-Oriented" Perspective: The authors argue that traditional machine learning models, which they term "Observation-Oriented," are limited in their ability to capture complex relationships and temporal dimensions. They propose a new "Relation-Oriented" perspective aimed at aligning machine learning models more closely with human cognition.

2. Relation-Defined Representation Learning Method: To implement this new perspective, the authors introduce a methodology called "relation-defined representation learning." This method is designed to autonomously identify and represent complex relationships across observational, temporal, and hyper-dimensional spaces.

The paper also provides examples from computer vision and health informatics to illustrate the limitations of existing models and to demonstrate the need for this new approach.

**Audience:**

Yes

**Broader Impact Concerns:**

I have no concerns about the ethical implications of this work that would require adding to a Broader Impact Statement.

**Claims And Evidence:**

No

**Requested Changes:**

### Critical for Acceptance:
1. **Experimental Validation**: The paper discusses plans for "extensive experimental validation" but does not present any results. This is crucial for substantiating the theoretical claims and should be included to secure a recommendation for acceptance.

2. **Clarify Claims**: The paper makes strong claims about the limitations of existing models and the revolutionary nature of the proposed approach. These claims need to be substantiated with evidence or toned down to avoid over-claiming.
### Would Strengthen the Work:
1. **Simplify Language**: The paper uses complex and academic language that could be simplified for broader accessibility. This would make the paper more impactful and easier to understand.

2. **Additional Examples**: While the paper does use examples from computer vision and health informatics, adding more diverse examples could strengthen the argument and make the paper more relatable to a broader audience.

3. **Discussion of Limitations**: A section discussing the limitations of the proposed "Relation-Oriented" perspective and "relation-defined representation learning method" would add depth to the paper and make it more balanced.

4. **Comparison with Existing Work**: A more detailed comparison with existing models and methods would provide context and help the reader understand the novelty and significance of the proposed approach.

5. **User-Friendly Summary**: A summary that distills the key contributions and findings into layman's terms could make the paper more accessible and help in conveying the importance of the work to a broader audience.

**Strengths And Weaknesses:**

### Strengths:
1. **Novel Perspective**: The paper introduces a "Relation-Oriented" perspective to machine learning, aiming to address the limitations of traditional "Observation-Oriented" models.

2. **Real-world Examples**: The paper uses examples from computer vision and health informatics to illustrate the limitations of existing models.

3. **Methodological Contribution**: The authors propose a "relation-defined representation learning method" that aims to identify and represent relationships across various dimensions.

### Weaknesses:
1. **Over-Claiming**: The paper makes extremely strong claims about the limitations of existing models and the advantages of their approach. These claims are not backed up with solid evidence to be convincing.

2. **Lack of Experimental Validation**: The paper mentions plans for "extensive experimental validation," but it's unclear whether these have been carried out. This is crucial for substantiating the claims made.

3. **Complexity of Language**: The paper uses complex and overly pompous language that makes it unaccessible to the relevant audience. Simplifying the language would make the paper more impactful, and allow readers to better judge the merit of the arguments made and the evidence that supports them.


Overall, the paper presents an intriguing new perspective but would benefit from clearer explanations, empirical validation, and a more cautious approach to claiming advantages over existing methods.

---

> ### Author Response · Authors · 2023-09-10
>
> Dear Reviewer jFmP,
>
> I am deeply grateful for your thorough review and insightful understanding of this paper. I am especially honored by your assessment of the proposed approach as having a “***revolutionary*** nature” – such a commendation is truly unparalleled in my experience.
>
> $\\ $
>
> **Regarding Experimental Validation:**
>
> $\ \ \ \ $ May I kindly request references to any published experiments that validate the basic assumptions of causal inferences$^{[1]}$ (i.e., Sufficiency, Faithfulness, Markov)? With such references, I can design tests for my causality definition without these assumptions.
>
> $\\ $
>
> **Regarding Language Complexity:**
>
> $\ \ \ \ $ Contrarily, I argue that this paper simplifies causal inference language. For instance, the intricacy of do-calculus is well-known, to the extent that entire papers are dedicated to dissecting the original do-calculus paper$^{[2][3]}$, while my reformulation of do-calculus spans merely half a page in this paper.
>
> $\ \ \ \ $ Could you pinpoint any instances within my paper, and contrast them with those from other publications, where I've used more complex language for the same idea? If the ideas I present are unprecedented, on what basis can we judge the originality of the phrasing as overly complex? Alternatively, you could critique the presented do-calculus reformulation.
>
> $\\ $
>
> **Regarding the Perception of Over-Claiming:**
>
> $\ \ \ \ $ This paper critiques the dominant modeling paradigm, as endorsed by Schölkopf$^{[4]}$, without asserting that the proposed method is the singular “correct” one in contrast to all existing others. For example, issues like those in Figure 8 are inherent under the current paradigm, making them inconspicuous irrespective of specific method choices or implementations. The intention behind the proposed approach is to demonstrate the viability of the new paradigm, emphasizing it is functional for practical use.
>
> $\\ $
>
> I genuinely value the insightful feedback from your review, and in response, I propose the following amendments:
>
> 1. I will incorporate more examples to underscore that the challenges presented in Figure 8 are pervasive across various domains, not just limited to health informatics.
> 2. In instances where I've referred to “existing methods”, I'll transition to using “prevailing paradigm” for clarity. As per your suggestion, I'll also add a concise summary to delineate the scope of my claims.
> 3. I plan to refine the term “extensive experimental validation” to “extensive functional experimentation” or “feasibility assessment” for better precision.
> 4. While I have made concerted efforts to minimize philosophical jargon, I concur that some terminology can be simplified for wider accessibility. However, I’d like to emphasize that as Ph.D.s, the essence of “P” inherently permeates our discourse and can't always be entirely sidestepped.
>
> $\\ $
>
> Regarding “Discussion of Limitations” and “Experimental Validation”, I believe these are more suited for new modeling methods than for introducing a new paradigm.
>
> A paradigm is to set up a viewpoint, determining what can be seen and what cannot.
>
> That is why concepts like Causal Sufficiency can be assumed without practical validation – We need this pair of eyes for seeing, although the eyes cannot see themselves, i.e., one cannot validate things that are prerequisites for validation (for your reference, this analogy is by Ludwig Wittgenstein).
>
> The significance of a ***revolutionary*** paradigm is in offering alternative viewpoints, rather than replacing the current one. Think about the potential future of AI - If some new issues can only be seen through the lens of quantum mechanics, why shall we remain our thought template at Newtonian mechanics? Moreover, the advent of relativity was never meant to erase Newtonian principles from physics textbooks.
>
> I eagerly await any additional feedback you might have.
> Thank you once again!
>
> Regards
>
> $\\ $
>
> [1] Stone, Richard. "The assumptions on which causal inferences rest." Journal of the Royal Statistical Society Series B: Statistical Methodology 55.2 (1993): 455-466.
> [2] Huang, Yimin, and Marco Valtorta. "Pearl's calculus of intervention is complete." arXiv preprint arXiv:1206.6831 (2012).
> [3] Heiss, Andrew. Do-calculus Adventures! Exploring the Three Rules of Do-calculus in Plain Language and Deriving the Backdoor Adjustment Formula by Hand | Andrew Heiss. 7 Sept. 2021, www.andrewheiss.com/blog/2021/09/07/do-calculus-backdoors.
> [4] Schölkopf, Bernhard, et al. "Toward causal representation learning." Proceedings of the IEEE 109.5 (2021): 612-634.

---

### Review · Reviewer_Jqrm · 2023-09-10

**Summary Of Contributions:**

Unclear.

**Audience:**

No

**Broader Impact Concerns:**

Unclear.

**Claims And Evidence:**

No

**Requested Changes:**

I urge the authors to streamline the text: focus the message, build a coherent and well-structured narrative, explain all terms used, remove all unnecessary observations and figures.

**Strengths And Weaknesses:**

**Strengths**

Unclear.

**Weaknesses**

- The text is *very* hard to follow.  The narrative keeps jumping from one
  topic to the next without providing any link, to the point it makes it hard
  to figure out what is being proposed exactly.

Notions are used without being introduced.  For instance, what is the
difference between observational, temporal, and hyperdimensional space?  Figure
1 provides very little guidance (the other figures are similarly confusing).
What is the difference between static temporal and dynamic temporal?  What is a
nonlinear treatment of time?  What is the link between these and the
"unrealistic hands" in the example?  And between this and causality?

The text is unnecessarily verbose, which combined with how difficult it is
to follow, makes it very frustrating.

The text needs substantial rework - specifically, to disentangle the various
messages.  I ended up having to skip Section 2 because of how messy it is.

**I do not plan to review the paper further unless it is given some kind of focus first.**

There may be useful - or even groundbreaking - messages in the conceptual
portion of the paper, but it is essentially impossible to extract them.

- The "lemmas" are not lemmas.  They are not formal mathematical statements and
  they have no proof.  I strongly suggest the authors to convert them into
  observations or claims.

- VAEs are *not* capable of disentanglement unless provided supervision on the
  latent representations or a strong architectural bias, which they normally do
  not have (see the works by Locatello and, separately, Hyvarinen).  This
  should be clarified in the text.

---

> ### Author Response · Authors · 2023-09-10
>
> No worries.
> I believe the insights from both the 1st and 2nd reviewers have indeed provided comprehensive evaluations of the paper's keynotes and theoretical foundations, as well as suggestions for revision.
>
> As you assess, "there may be groundbreaking." When new and groundbreaking ideas emerge, they might not be immediately integrated with established knowledge.
> Not mean to flatter myself, but if physics teachers were invited to review the relativity paper Einstein submitted, the same situation would happen.
>
> I'd like to point out that all your questioned concepts are illustrated by the "unnecessary observations and figures". And thanks for emphasizing the conditions for VAEs' capabilities, which I've taken as a given, considering their widespread understanding.
>
> Thank you for dedicating your time to this review.

---

> ### Author Response · Authors · 2023-09-15
>
> Dear Reviewer Jqrm,
>
> Upon reflection, I've come to realize the invaluable insights your comments provided. I sincerely apologize for the issues in the initial version and hope this revised manuscript will offer a smoother review experience.
>
> I eagerly await your feedback, and thank you for your understanding.
>
> Best regards,

---

### Review · Reviewer_Fvjm · 2023-09-14

**Summary Of Contributions:**

I can see that a lot of effort went into this paper, so I'm reluctant to be this harsh, but honestly this paper is so confusingly written that I don't know that I can list the the contributions. It jumps from issues in generative models (failures to correctly generate hands), to causality and then attempts to claim that these are both AI alignment failures: but then claims that AI alignment is "encapsulated by the essential question, 'Why are some relationships unseen to AI?'". This is both an incorrect definition of alignment (which studies the mismatch between proxy rewards and humans' intended goals), and even if we take this definition as correct, this relationship is not clearly defined.

**Audience:**

No

**Claims And Evidence:**

No

**Requested Changes:**

Honestly - this paper needs a significant rewrite to clarify what it's actually trying to claim. It seems to change its mind over the duration of the paper: the first couple of pages spend significant amount of time on AI producing mangled fingers, but then hands are never mentioned again after page 4 and the connection with causality is never explained. Similarly references to AI alignment seem to totally misunderstand the problem that AI alignment is aiming to solve.

**Strengths And Weaknesses:**

Strengths:
 * The topics that the paper studies - causality, relational models, abstraction - are all interesting and worthy of study.
 * The DAGs with time of effect explicitly represented time and the world line-style diagrams are potentially interesting if fleshed out properly. If I were supervising this project, I would encourage the author(s) to scope narrowly to that idea and explain precisely where it is useful. But this would entail a significant overhaul that goes beyond the TMLR review process. The major issue with the current state of this paper is there are far too many threads of ideas that are never connected or precisely explained.

Weakness:
 * There are so many unsupported and incorrect claims. I will list examples below:
1. "conventional models $f(\cdot)$ are limited to linear relationships with respect to $t$" - which models? RNNs, Transformers, time series models with appropriate basis function expansions, etc. all allow nonlinearity.
2. "Go gaming" as an example of an observational learning task: in Go, you're learning a policy which is inherently interventional (via RL).
3. No citations for issues with diffusion models and hands (yes I've seen the social media posts, but these should be cited, or better you should cite a paper that has studied this issues more rigorously).
4. "AI's capability within the temporal dimension remains notably constrained" (bottom of page 1). How is it constrained? Much of the recent progress in AI has been in modelling sequential data (e.g. WaveNets for audio, Transformers for Text, Video-based model, etc.). When we have temporal observations at sufficient time resolution, we have excellent models for sequential problems. The challenge is typically in collecting the right data.
5. The "Division of cognitive space" does not formally define the three categories, so it is not clear how problems are categorized. CNNs are given as an example of a model in observational space, as is a quadrotor's location - but a moves over time, so surely it's in temporal space? And if not, what is the formal distinction?
6. The "hierarchy of observational features" section confuses the difficulties associated with generation from those associated with recognition. The section highlights how humans hierarchically decompose images (this claim should be supported with citations) when *recognizing* hands, and contrasts it with the failure of diffusion models to generate accurate hands. But generation is a far harder problem than recognition - so even though I agree that it would be nice to have hierarchical decompositions, these examples do not imply that. E.g. If regular (non-artist) people were to draw hands, they would be equally unrealistic not because people don't know what a hand looks like, but because drawing (generation) is far harder than recognition (for an example - see Lawson 2006 "The science of cycology: Failures to understand how everyday objects work" for examples of how people can't draw functional bicycles from memory).
7. Page 5 paragraph 2 describes how "traditional medical effect estimation" is conducted (and how it is misguided) - but the described method looks nothing like textbook causal effect estimation which is used in epidemiology, and no citations are provided to show where this approach is used. So as it stands, the argument is essentially a straw man.
8. Page 5 "The elusive hidden confounder" makes one useful---though well understood (see e.g. Athey et al 2019, The Surrogate Index)---observation: if our measures are taken too early we may not see the full causal effect. But it confuses the role of hidden confounders. We do not introduce hidden confounders to "enhance human understanding", hidden confounders appear in a DAG to acknowledge the factors that we couldn't measure even if we would have liked to. So while it is true the a patient's state is in principle observable at any time, it is seldom the case that one has continuous observations of this state (you'd need medical devices on your subjects 24/7 to achieve this!), so causal inference needs to address the fact that observations are limited, and in some cases these limitations result in hidden confounding which requires difference inference approaches.
9. Page 5: bottom paragraph. The paper states, "Curiously, one might find it rare to see “incorporation of time” defined as the distinctive factor between causality and mere correlation in causal inference theories." - this is because the incorporation of time is not sufficient to distinguish between causal and correlational relationships. See Peters et al. Elements of Causal Inference, chapter 10.

* Lemma 1 is both not a lemma (it's a just a statement, not a formal proposition), and also doesn't seem to be correct (or at least "incorrect" unless there's a missing formal definition of "dynamical" that make it correct). The statement "Correlation is connection between features, which are not dynamical." misses that time-independent correlation between two variables can be causal if there is no confounding: e.g. in a well-designed randomized control trial, any correlation is causal.

* Lemma 2 is a definition not a lemma.

---

> ### Author Response · Authors · 2023-09-15
>
> Deer Reviewer Fvjm,
>
> I apologize for the one-day delay in updating the version following your review. The manuscript has been extensively streamlined, and I trust that this revised version will provide a more seamless review experience.
>
> Thank you very much for your dedication and understanding.

---

> ### Author Response · Authors · 2023-09-16
>
> Dear Reviewer Fvjm,
>
> For your convenience, I'll address your concerns based on the previous version, and indicate where the answers can be found in the updated rewrite.
>
> 1. All models. It seems you haven't recognized that this isn't the regular "nonlinearity" you referred to. Please see the 2nd paragraph of the Introduction in the updated version.
>
> 2. Yes, but that "interventional" is on a singular, absolute timeline, not the real "temporal dimension". Please read section 1.3 and 4 in the updated version.
>
> 3. Good suggestion and I'll consider it. I didn't cite any because haven't found any work that pinpointed "unobservable hierarchy" as the core, but all delved into phenomenological descriptions.
>
> 4. Once more, what you perceive as the "temporal dimension" is, in fact, just "a timeline." Please review the entire updated version. It may shed light on some fundamental questions you haven't considered before.
>
> 5. Please read section 1 of the updated version. And the answer is no, it's not in a temporal space.
>
> 6. Thanks for the information, but it's not the objective. The extensive descriptions in the previous version may have misled readers. I apologize for that oversight. Please refer to the updated Section 2.1, which is now concise and spans only half a page.
>
> 7. You're correct in suggesting that broader applications should be included. Accordingly, this has been addressed in the updated version 2.2.
>
> 8. Sorry, but your original statement, "hidden confounders appear in a DAG to acknowledge the factors that we couldn't measure even if we would have liked to" exactly means "enhance human understanding". Because you are the one who read this DAG, not your model.
>
> 9. Clearly, my focus is on "Why isn't it sufficient? And why should we differentiate between causality and correlation?" I'd like to remind you that for a researcher, a textbook is not the final authority.
>
> I've updated "Lemma" to "Theorem" — thank you for pointing that out. I suggest first distinguishing between "a timeline" and "the temporal dimension." Once that's clear, the theorems should be more understandable for you.
>
> The streamlined current version provides a clearer perspective on the connections behind all the examples. I encourage you to review it. Thank you again for your invaluable insights on the previous draft.

---

### Author Response · Authors · 2023-08-31
**Open to direct advancement to "rebuttal and discussion" stage, if limited reviews**

Dear Editors and Reviewers,

I'd like to extend my sincere appreciation for dedicating your time and expertise to review this paper, especially given its length and deep physiological topics it delves into.

This paper aims to challenge the prevailing model paradigm $Y_{t+1}=f(X_t)$ that has dominated for decades - With specified $t$-value as a timeline attribute, we cannot truly incorporate "time" as a computational dimension.
An illustrative example is presented in Figure 8, where existing methods struggle to avoid inherent violations.

Introducing the concept of “temporal distribution” can, in fact, simplify many issues, e.g., the identifiable difficulties in causal learning, the do-calculus insights, the “unseen” aspects of human knowledge to AI, etc.

$\ $

Imagine there are 100 rich “mines” in the grand scheme of AI applications, while presently, we've predominantly focused on just 2 - languages and images – both not involving the dimensionality of time. My aim with this paper is to present a potential key to steer interest towards the remaining 98 mines.

A looming AI misalignment crisis could arise from the creation of Gödel Machines (in language and image domains) that possess the capability to rewrite but lack a genuine understanding of the knowledge they encounter. The relation-oriented principle proposed herein could serve as a conduit to impart causal reasoning to AI.

$\ $

Given the unique nature of this paper, direct communication may prove more effective for clarification during the review process. Should there be fewer than three reviews provided, I am amenable to moving directly to the "rebuttal and discussion" stage. Your consideration is deeply appreciated.

Thank you!

---

### Author Response · Authors · 2023-09-15

Dear Reviewers,

The updated version has been extensively revised in response to all the comments provided. The manuscript is now five pages shorter.

I am genuinely looking forward to your feedback, and I sincerely apologize for the shortcomings in the initial version.

Thank you once again for your dedication and time!

---

### Author Response · Authors · 2023-09-25

Dear Reviewers and Editors,

I hope you had a pleasant weekend.

I would like to extend my heartfelt gratitude for your constructive feedback thus far, particularly to Reviewer Jqrm and jFmP. Thanks to your insights, the updated version of the manuscript has significantly improved in terms of organization and readability.

I eagerly await any comments on the current version, as well as any additional responses related to ongoing discussions.

Wishing you a wonderful day ahead!

Best regards,

The author

---

### Author Response · Authors · 2023-09-27

Dear Reviewers and Editors,

I hope all is going well for you. Wishing you a pleasant remainder of the week.

Best,

---

### Author Response · Authors · 2023-09-29

Dear Reviewers and Editors,

Wishing you a pleasant Friday and a wonderful weekend ahead.

Best,

---

### Author Response · Authors · 2023-10-01

Dear Reviewers and Editors,

Hope your weekend is going well, and I am looking forward to any informative rebuttals you may have.

Best regards,

---

### Author Response · Authors · 2023-10-15

Dear Chief and Editors,

The final version was uploaded on October 14th.

I've streamlined the content, removing all complex metaphysical descriptions, observations, and analogies, and focused solely on computer science language.

If any part of the manuscript appears unclear or contains potentially inappropriate claims, please pinpoint the location, and I'll make the necessary adjustments.

I eagerly await your feedback and decision.

Best,

---

### Decision · Action_Editor_iQDd · 2023-10-28

**Recommendation:** Reject

**Comment:**

The topic of the paper touches many important limitations of the current landscape of AI, at the intersection of generative causal models, representation learning and alignment. In its current form, however, it is hard to pinpoint a set of crisp contributions from this work, and therefore evaluating them is not easy.

Specifically, the claims advanced are too fuzzy, worded in a non precise way and not backed up by precise and concrete evidence. Reviewers highlighted these aspects and agreed that a major rewriting was necessary.

While a revised version improved the presentation, being shorter and providing somehow more precise definitions, all reviewers agreed that in this current state the paper is not ready for publication.

**Audience:**

The topic of causality is definitely aligned with the scope of TMLR. The current version of the manuscript, however, fails to convey how it contributes to advance the field of causal ML in a precise and rigorous way, therefore making it harder for the TMLR audience to get its message.

**Claims And Evidence:**

As highlighted by all reviewers, the claims in this work are not rigorous and not substantiated enough to recommend acceptance.